# Liquid metal-based synthesis of high performance monolayer SnS piezoelectric nanogenerators

Hareem Khan[1], Nasir Mahmood [1], Ali Zavabeti [1,2], Aaron Elbourne [3], Md. Ataur Rahman[1], Bao Yue Zhang [1], Vaishnavi Krishnamurthi[1], Paul Atkin[1], Mohammad B. Ghasemian [4], Jiong Yang[4], Guolin Zheng[3], Anil R. Ravindran[1,5], Sumeet Walia [1,6], Lan Wang[3], Salvy P. Russo[3], Torben Daeneke [1], Yongxiang Li[1✉] & Kourosh Kalantar-Zadeh [4✉]

The predicted strong piezoelectricity for monolayers of group IV monochalcogenides, together with their inherent flexibility, makes them likely candidates for developing flexible nanogenerators. Within this group, SnS is a potential choice for such nanogenerators due to its favourable semiconducting properties. To date, access to large-area and highly crystalline monolayer SnS has been challenging due to the presence of strong inter-layer interactions by the lone-pair electrons of S. Here we report single crystal across-the-plane and large-area monolayer SnS synthesis using a liquid metal-based technique. The characterisations confirm the formation of atomically thin SnS with a remarkable carrier mobility of ~35 cm$^2$ V$^{-1}$ s$^{-1}$ and piezoelectric coefficient of ~26 pm V$^{-1}$. Piezoelectric nanogenerators fabricated using the SnS monolayers demonstrate a peak output voltage of ~150 mV at 0.7% strain. The stable and flexible monolayer SnS can be implemented into a variety of systems for efficient energy harvesting.

[1] School of Engineering, RMIT University, Melbourne, VIC 3000, Australia. [2] Department of Chemical Engineering, The University of Melbourne, Parkville, VIC 3010, Australia. [3] School of Sciences, RMIT University, Melbourne, VIC 3000, Australia. [4] School of Chemical Engineering, University of New South Wales (UNSW), Kensington, NSW 2052, Australia. [5] Institute for Frontier Materials, Deakin University, Waurn Ponds, VIC 3216, Australia. [6] Functional Materials and Microsystems Research Group and the Micro Nano Research Facility, RMIT University, Melbourne, VIC 3001, Australia. ✉email: yongxiang.li@rmit.edu.au; k.kalantar-zadeh@unsw.edu.au

Flexible and soft self-powered sources for harvesting energy from mechanical stimuli in the environment constitute the burgeoning field of nanoenergy[1–4]. For such applications, limitations of conventional piezoelectric ceramic thin-films arise due to their brittle nature[5]. In contrast, the distinctive physical properties of two-dimensional (2D) materials provide viable avenues in this realm[1,6–8]. The planar integrity of 2D materials, their ability to withstand large strains and the emergence of piezoelectricity in specific 2D crystals are the unique traits for developing nanogenerators[9]. However, technical limitations of forming 2D coatings of strong piezoelectricity, on large scales, have so far hindered their application in actual devices. Additionally, demands in forming high quality and consistent monolayers in order to gain strong output voltages, and reduced grain boundaries across such layers, to increase the current, have not been met. More importantly, accessible output voltages for so far explored 2D piezoelectric materials have not provided satisfactory values that show the prospect of future industrial translations.

One particular group of 2D compounds, the group IV monochalcogenides has been theoretically identified as offering strong intrinsic piezoelectricity[5], and if synthesised can offer large output voltages. Specifically, tin monosulphide (SnS) has been predicted, using density functional theory (DFT) calculations, to present a very high piezoelectric coefficient; relaxed-ion calculations as $d_{11} = \sim 144$ pmV$^{-1}$ and clamped-ion as $d_{11} = \sim 22$ pmV$^{-1}$ and hence a remarkable mechanical-to-electrical conversion efficiency is expected. High piezoelectric coefficients in group IV monochalcogenides are suggested to appear as a result of their puckered structure, generating in large in-plane polarisation induced by an applied stress[10,11]. The puckered $C_{2v}$ crystal lattice, together with soft bonds, are accountable for the large increase in the magnitude of the piezoelectric coefficient[10]. In addition, SnS has semiconducting properties more suitable than many other 2D materials for electronics and energy harvesting. It has been shown that bulk SnS crystal offers a bandgap of ~1.25 eV, comparable to that of silicon, with conduction and valence band edges located at −3.8 and −5 eV, respectively[5,12]. Additionally, both measurements and DFT calculations suggest that monolayer SnS does not open its bandgap with reference to that of its bulk[5,13]. Therefore, the incorporation of the monolayer SnS into flexible and wearable piezoelectric nanogenerators (PENG) should be of greatest potential[1].

Despite the predictions of large piezoelectricity and hence ability to achieve the high conversion efficiency for nanogenerators made from monolayers of group IV monochalcogenides, their synthesis has so far been mostly limited to small scale surface coverage due to significant challenges, with a trade-off between lateral size and thickness and limited sliding effect between layers[14–16]. Conventional 2D exfoliation synthesis techniques, have so far failed to reach wafer scale dimensions, which has hindered the application of such materials and their oxides[17,18]. Mechanical exfoliation methods only result in 2D batches of small lateral dimensions on substrates[16]. Chemical and physical vapour deposition techniques, often claiming to be suitable wafer scale synthesis while forming films contain high frequencies of grain boundaries which may lead to diminished electronic properties[19]. Similarly, utilising liquid phase exfoliation methods only results in films of stacked nanosheets with small lateral dimensions, rendering them impractical for energy harvesting applications that rely on stress upon or along 2D sheets[16,20]. To make the matter more challenging, the formation of high quality monolayers of group IV monochalcogenides, specifically SnS, using conventional exfoliation, has been suggested to be limited due to the strong inter-layer interactions by the lone-pair electrons of S, which are much stronger than the van der Waals forces between the layers[17,20].

Here we present a synthesis technique for achieving stable, large area and homogenous monolayer SnS sheets for wafer scale processes. The technique relies on the self-limiting formation of the large area tin sulphide on the surface of molten tin[21–23]. The reaction follows the Cabrera-Mott mechanism[24], which elucidates the establishment of ultrathin interfacial layers with finite thicknesses of a single or few monolayers on elemental metals. The use of a liquid growth substrate and the high surface tension of liquid metals ensures a near ideal growth interface, leading to a product with minimal grain boundaries and dislocations[25]. The grown interfacial sheets can then be readily delaminated from the liquid metal interface as they impose minimal forces on the liquid interface due to its lack polarity or differently described an overall reduced van der Waals force imposed onto the layer[21,22]. Utilisation of a direct surface reaction for the synthesis enables the production of stable superior quality monolayer tin compounds with no competing impure species in the immediate environment that is used for developing PENG of exceptional performance.

## Results

**Synthesis and characterisation of the monolayer SnS.** Previously, the synthesis of monolayer group IV monochalcogenides has proven to be challenging due to strong layer intercoupling[17,20]. A summary of the compounds and the methods used to make them to date are summarised in Supplementary Table 1. In our experiments, to synthesise the monolayer of SnS, we exposed a molten droplet of Sn to an anoxic atmosphere containing a sulphur source (50 ppm H$_2$S gas in N$_2$ background) at 350 °C in a custom-made setup (Fig. 1a and Supplementary Fig. 1). The surface forms a sulphide skin in a self-limiting Cabrera-Mott reaction. The ultrathin SnS sheets on the templating ultra-smooth surface of liquid metal are then exfoliated and transferred onto a desired substrate via van der Waals adhesion.

Our morphological study shows the formation of ultrathin nanosheets featuring large lateral dimensions (Fig. 1b). Using optical microscopy, the layers were identified (Supplementary Fig. 2) as millimetre to centimetre-size delaminated sheets. Further analysis by high-resolution scanning electron microscopy (Supplementary Fig. 3) shows continuous laterally large sheets with no pinholes. While the size of the monolayer SnS can be potentially very large, in this work, we consistently used an average droplet size of 1 cm in diameter and a 45° angle of contact resulting in monolayer sheets of hundreds of microns in dimensions (Supplementary Note 1 and Supplementary Fig. 4). The mechanical exfoliation parameters were used diligently to obtain consistent results and were the most suitable parameters for the custom-designed setup we used for the synthesis (taking into account the size of the heater and workspace area in the glovebox) but are easily scalable to any setup. Controlling the sulphur concentration in the environment was critical in achieving thin pure-phased monolayer SnS (Fig. 1c). During the optimisation process, we noted that exposing the liquid metal for an extended time could yield islands of SnS$_2$ on SnS monolayers. Similarly, the emergence of stoichiometric SnO$_2$ islands on SnO sheets has been reported when exposing liquid tin to oxygen[21]. A growth map illustrating the link between growth conditions and obtaining SnS or SnS$_2$ products is presented in Supplementary Information (Supplementary Note 2 and Supplementary Fig. 5). Selected-area electron diffraction (SAED—inset Fig. 1c) indicates the high crystallinity of the sheets.

The near single crystallinity, at least within micron domains, and low concentration of grain boundaries across the planes are also identified using dark field transmission electron microscopy (TEM) (Fig. 1d). We conducted SAED measurements across various locations on the synthesised/transferred monolayers of

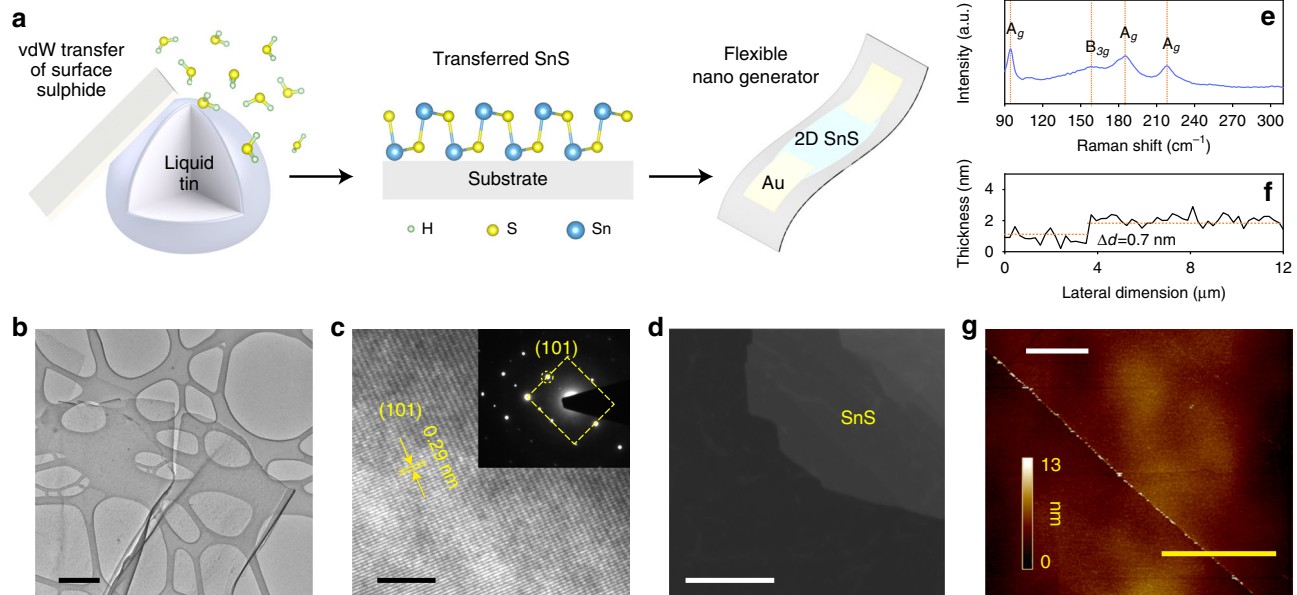

**Fig. 1 Synthesis schematic representation and material characterisations of the 2D SnS. a** Schematic depiction of the synthesis process, monolayer of SnS and its application on a nanogenerator transducer. **b** TEM image of monolayered SnS nanosheet. Scale bar is 500 nm. **c** HRTEM fringe pattern showing *d*-spacing of 0.29 nm matched to the plane (101) with inset that shows SAED pattern also indicating a 0.29 nm *d*-spacing matched to the (101) plane. Scale bar is 5 nm. **d** Dark field TEM image of a 2D SnS that shows very low presence of grain boundaries, confirming near single crystallinity of the planes. Scale bar is 100 nm. **e** Raman spectrum showing the four characteristic Raman active vibration modes that are associated with the A$_g$ modes and the B$_{3g}$ mode. Four characteristic Raman active modes at 96, 190 and 220 cm$^{-1}$ are associated with the A$_g$ modes, while a peak at 163 cm$^{-1}$ assigned to its B$_{3g}$ mode are expected in bulk SnS. **f** Thickness profile showing sheet thickness of ~0.7 nm, which can be matched to one-unit cell thickness of monolayer SnS and **g**) the corresponding AFM of the SnS monolayer (Thickness profile in **f** was taken along the yellow line). Scale bar is 8 μm.

SnS which confirmed consistent crystallographic orientations, suggesting the single crystalline characteristic (Supplementary Note 3 and Supplementary Fig. 6). From high resolution TEM (HRTEM—Fig. 1c), we obtain a *d*-spacing of 0.29 nm that can be matched to (101) plane of orthorhombic SnS.

Raman spectroscopy was utilised to confirm the synthesis of ultra-thin SnS (Fig. 1e). The observed spectrum featured a slight red shift as well as peak broadening which is expected due to the photon confinement effect that arises in monolayer SnS[20,26] X-ray diffraction (XRD) of multiple SnS prints re-confirms the growth of orthorhombic SnS, matching PDF card No. 39-0354 (Supplementary Fig. 7).

The monolayer nature of the sheets was further confirmed by atomic force microscopy (AFM), showing a thickness of one unit-cell of SnS on the majority of the substrate's surface with the highly homogenous resolvable thickness of ~0.7 nm (Fig. 1f, g). Supplementary Figure 8 shows a bilayer AFM with a measured thickness profile of ~1.7 nm, presented in the inset, which is equivalent to twice the unit cell thickness.

The X-ray photoelectron spectroscopy (XPS) of SnS shows core levels of Sn and S that confirms high purity of the as-synthesised SnS (Supplementary Fig. 9). The deconvoluted Sn 3*d* spectrum shows two peaks at the binding energies of 485.9 and 494.3 eV, associated with the 3*d*$_{5/2}$ and 3*d*$_{3/2}$ energy states, respectively, as the affirmation of Sn$^{2+}$ presence (Fig. 2a). The S 2*p* doublet is well resolved with spin-orbit component separation of 1.3 eV and with an area ratio of 1:2, corresponding to the 2*p*$_{1/2}$ (162.5 eV) and 2*p*$_{3/2}$ (161.2 eV) features, respectively (Fig. 2b, Supplementary Note 4 and Supplementary Fig. 10)[27].

The semiconducting properties of monolayer SnS were also studied. The XPS valence band indicates the energy difference between the Fermi level and the valence band maximum as ~0.6 eV (Fig. 2c). The photoelectron spectroscopy in air (PESA) plot (Fig. 2d) indicates the ionisation potential and hence the

Fermi level relative to the vacuum which is seen to be approximately −4.7 eV for monolayer SnS. The Tauc plot (Fig. 2e) was obtained from the UV-Visible spectrum indicating an optical bandgap of ~1.4 eV, which is well-matched to the predicted values in theoretical reports for monolayer SnS[5,13].

The homogeneity and reduced grain boundary of the as-synthesised semiconducting monolayer SnS sheets were remarkable. These were assessed by carrying out Hall-effect measurements (Supplementary Fig. 11). The carrier mobility and densities were ascertained as ~35 cm$^2$ V$^{-1}$ s$^{-1}$ and 1.33 × 10$^{11}$ cm$^{-2}$, respectively. The positive slope of the Hall coefficient validates the *p*-type semiconducting nature of the material (Fig. 2f) The hole mobility is significantly higher than previous reports for monolayer SnS[13], confirming reduced grain boundaries. The electronic band structure of the *p*-type semiconducting monolayer SnS is constructed as presented in Fig. 2g.

**Device fabrication and test**. The piezoelectric properties of the monolayer SnS were explored by developing piezoelectric nano-generators. SnS monolayer was delaminated onto the relevant flexible substrate of either atomically flat synthetic mica or polydimethylsiloxane (PDMS) as described in the "Methods" section. Scanning electron microscopy (SEM) image of the device is shown in Supplementary Fig. 12. The devices were then investigated by the application of mechanical strains in various modes.

Piezoresponse force microscopy (PFM) analysis was used for exploring the piezoelectric property of monolayer SnS as a complementary assessment. An out-of-plane coefficient is not expected due to its structure so the lateral PFM (LPFM) mode was employed to measure the in-plane coefficients. When using LPFM microscopy mode, the stimulus is provided from above the sheet (hence out of plane) and the response is measured in-plane. The calibration sample was measured in the exact same way.

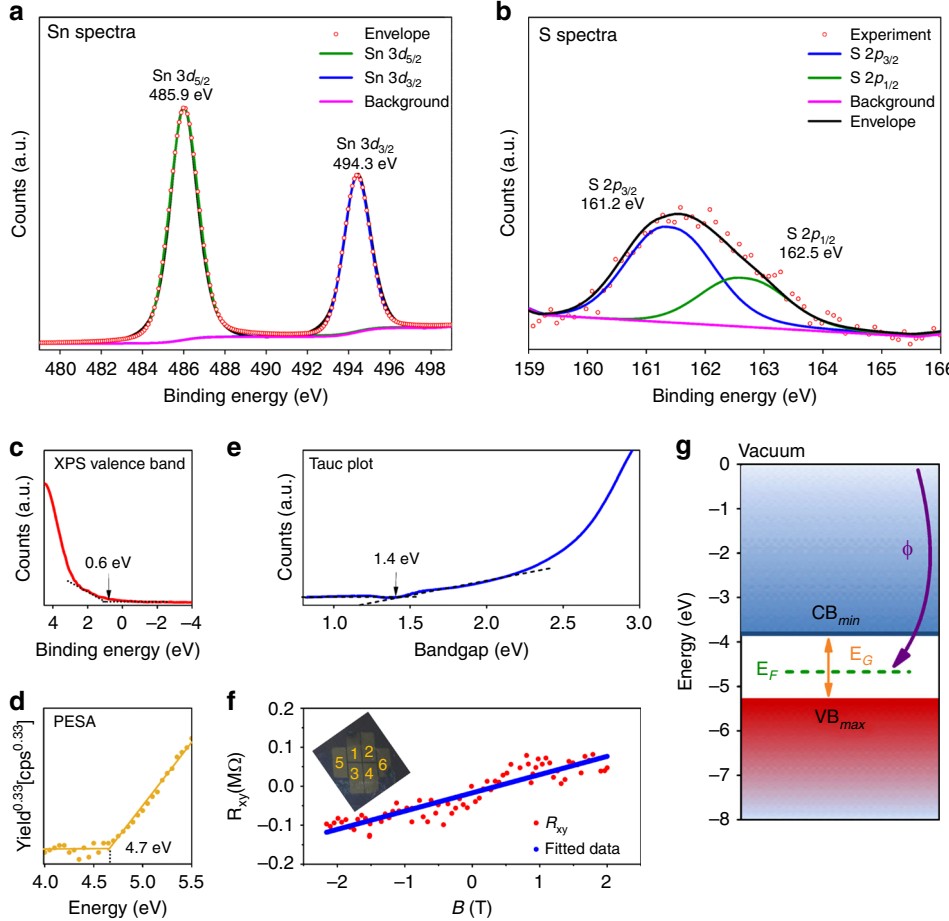

**Fig. 2 Compositional and electronic characterisations of the synthesised monolayer SnS. a** Sn 3d XPS spectra deconvoluted and fitted to show the typical peaks of $Sn^{2+}$ binding energies at 485.9 and 494.3 eV. **b** S 2p XPS spectra deconvoluted and fitted to show peaks at binding energies of 161.2 and 162.5 eV **c** XPS valence band spectrum indicating the energy difference between the Fermi level and the valence band maximum is ~0.6 eV. **d** PESA spectrum where it is observed that the Fermi energy level is ~4.7 eV below vacuum (cps-counts $s^{-1}$). **e** Tauc plot converted from the UV-Visible spectroscopy indicates the optical bandgap as ~1.4 eV. **f** Hall resistance ($R_{xy}$) of the monolayer SnS as a function of the applied magnetic field (B) with optical image of six-terminal Hall device in the inset. **g** Resulting assessed electronic band structure system of monolayer SnS confirming p-type semiconducting nature.

Kelvin probe force microscopy (KPFM) measurements were performed prior to the PFM measurements to cater for any electrostatic charge contributions to the output signal[28] and were performed again after the measurements to validate the data further (Supplementary Figs. 13 and 14). Figure 3a shows the topography of the synthesised SnS monolayer with a height profile of ~1 nm, confirming the monolayer nature of the films. An inset histogram also shows the height distribution across the scan. Figure 3b, c show a clear contrast between the SnS monolayer and the $SiO_2$/Si substrate when measuring the lateral amplitude and phase. Figure 3d shows the lateral piezoresponse as a function of the magnitude of the driving voltage that is applied to the SnS monolayer and the periodically poled lithium niobite (PPLN) test sample. The trend shows a distinct linear increase in the piezoresponse as applied voltage increases for both samples. This confirms the source is piezoelectric and therefore the slope can be used for calculating the effective piezoelectric coefficient. The plot is fitted to a linear line, which gives 1.2558 and 0.2864 a.u. $V^{-1}$ for the monolayer SnS and the PPLN sample, respectively. Because $d_{eff} = 5.95$ pm$V^{-1}$ for the PPLN in the lateral direction (Supplementary Fig. 15), an effective piezo-electric coefficient of $26.1 \pm 0.3$ pm$V^{-1}$ is obtained for monolayer SnS. These PFM results confirm that the synthesised SnS monolayer shows a relatively strong in-plane piezoelectricity. PFM measurement setup is shown in Supplementary Fig. 16.

Two modes of testing were used for the devices, one to test the $d_{31}$ mode, whilst the other to test the $d_{11}$ mode. The first mode of testing was the $d_{31}$ mode, where force was applied normal to the-plane direction by tapping the device surface in a controlled manner with a measured peak-to-peak load amplitude of ~4 N and in the frequency range of 1–10 Hz (Supplementary Fig. 17). This test was conducted to explore viability of embedding such a nanogenerator in low-frequency applications, where force is applied normal to the surface such as walkways catering for commuters or streets for cars. For these tests both double electrode and multiple electrode devices were fabricated (see Methods).

Devices in two-electrode and multi-electrode configurations were fabricated to test the $d_{31}$ mode. The optical image of two-electrode device is presented in Fig. 4a and this device which was exposed to tapping force of 3 Hz generated voltages in excess of ~150 mV peak value (~300 mV peak-to-peak). The voltage spikes up as the force is applied and spikes down as the force is removed, a typical response for piezoelectric systems[29]. Figure 4b shows the voltage output of this two-electrode device which for this particular device shows an output of is ~160 mV peak value over an applied load resistance of 10 MΩ, at 3 Hz frequency. Figure 4c shows the voltage output of the multi-electrode device which is ~190 mV peak value over an applied load resistance of 10 MΩ, 5 Hz frequency. The device output was tested at different tapping frequencies ranging from 1 to 10 Hz (Fig. 4d).

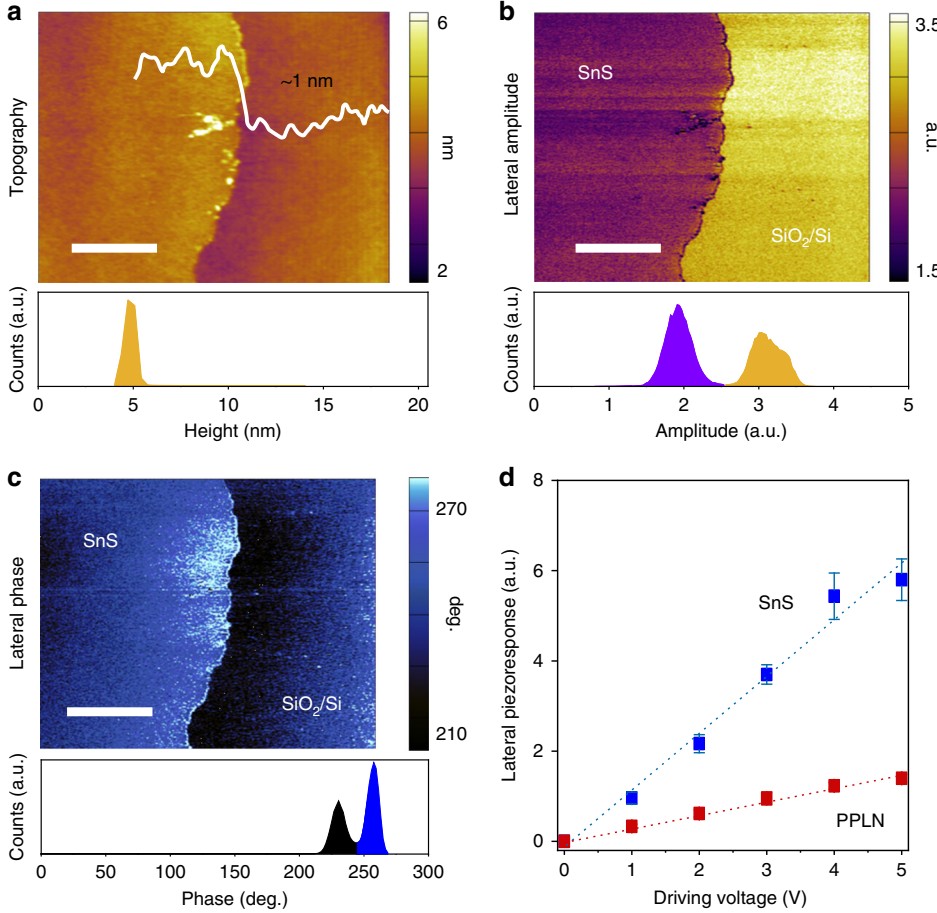

**Fig. 3 Lateral piezoelectric response of the monolayer SnS. a** Topography of the monolayer sheet with the height profile illustrating sheet thickness of ~1 nm, which is approximately a monolayer. This is also confirmed by the histogram showing the height distribution. **b** Lateral amplitude of the SnS monolayer with inset of histogram. **c** Lateral phase of the SnS monolayer with inset of histogram. **d** Lateral piezoresponse of monolayer SnS and PPLN vs the driving voltage. Scale bars are 2 μm. Error bars denote the standard deviation of the measurements.

The device which as such showed a high output voltage would be suitable for practical energy harvesting applications, which arise from randomly available mechanical stimuli in the ambient. The envisioned utility of such devices is for the energy harvesting of naturally occurring excitation in the environment rather than a forced or regular driving force. We hence tested the nanogenerator in a range of frequencies. This suggests that this nanogenerator can be applied in scenarios with low-frequency stimulations such as biomedical sensors, or walkways in malls or airports. It was observed that the frequency of 5 Hz gave a maximum output value of ~190 mV on average, resulting in the maximum peak power output of 30.4 pW. Above this, the reason the signal is damped which is most likely due to the inability of the flexible substrate to respond at the excitation rate of the stimulus force being applied and hence the output voltage is in turn reduced. However, this can be easily overcome by better choice of flexible material for the device's substrate. Figure 4e shows the output voltage for a continuous applied force for ~4700 cycles at a frequency of 5 Hz. This uniform output confirms the mechanical durability of the device and indicates its ability to be utilised in practical applications, where it would need to work for long periods of time, and in harsh environments.

To show the viability of the developed flexible SnS monolayers in scavenging energy as a wearable nanogenerator (Fig. 5a and Supplementary Fig. 18) the device was tested in $d_{11}$ mode by applying a strain in a tensile arrangement with the piezoelectric sheet on the surface of the device. An average peak output voltage of ~150 mV (~300 mV peak-to-peak) was recorded upon approximated strain of 0.7%. Current measurements were also done with 1 GΩ passive resistance (Supplementary Fig. 19) which produced ~160 pA output current.

This generated voltage is significantly higher than any previously reported PENG output on monolayer-based nanogenerators such as $MoS_2$ and $WSe_2$[6,8,9,30]. Further tests to ensure the output was from the actual monolayer SnS included a test of the blank devices without any 2D material and with a variety of other input signals such as triangular and sine pulses and both tests confirmed the authenticity of the source of the device output (Supplementary Figs. 20 and 21). Both triboelectric and flexoelectric effects can be ruled out for the generation of the outputs. Triboelectric effect is ruled out after showing that devices with no materials between electrodes or devices with SnO between electrodes could not generate any voltage after applying mechanical stimuli (Supplementary Fig. 22). As for flexoelectric, this phenomenon is diminished in monolayer structures due to no strain gradient (Supplementary Fig. 23).

Considering that highest reported output voltage for $MoS_2$ and $WSe_2$ are in the order of 10–57 mV, our device has the output of at least 3 times larger than of the best of those reports[6,9]. This difference can be associated to the giant $d_{11}$ value for monolayer SnS as per DFT calculations in comparison to that of $MoS_2$ and $WSe_2$ that are <4 pmV$^{-1}$ in majority of past reports[7].

Power density of the devices was calculated by taking the ratio of the peak voltage power and the area of the devices. The highest

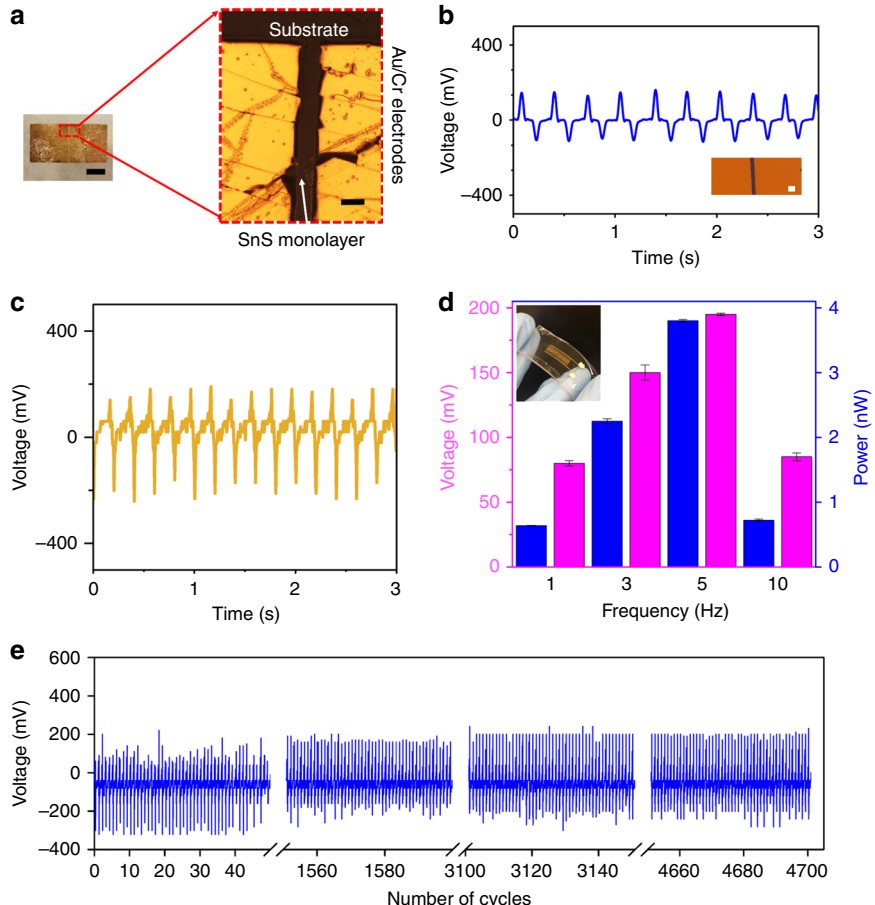

**Fig. 4 The outputs of the devices with synthesised monolayer SnS using tapping mode to excite $d_{31}$. a** The optical image of the two-electrode device. Scale bars are 1 mm and 50 μm for left and right images, respectively. **b** Response of the output voltage of the two-electrode device. Inset shows an optical image of the device. Scale bar is 50 μm. **c** Response of the output voltage of the multi-electrode device excited at 5 Hz. **d** A graph showing the variation of average output power and voltage at varying tapping frequencies and inset is the photo of the PDMS-encapsulated mica flexible nanogenerator (multi-electrode device). Error bars denote the standard deviation of the measurements. **e** Stability test of output voltage with stimulus applied at frequency 5 Hz for ~4700 cycles (multi-electrode device).

previous report on power density of 2D nanogenerator devices has been reported for MoS$_2$ (2 mWm$^{-2}$)[9], while our nanogenerator shows a significantly higher power density of ~24 mWm$^{-2}$. Figure 5b and Supplementary Table 2 summarises the parameters and compares our work with previous reports published on nanogenerators which are based on 2D materials highlighting their key performance criteria including the output voltage and power density.

## Discussion

In summary, synthesis of large and highly crystalline semiconducting monolayer SnS was shown via the van der Waals exfoliation technique from a liquid metal surface of tin melt in the H$_2$S environment. A comprehensive set of characterisations confirm the mono-crystallinity, p-type nature, high carrier mobility of ~35 cm$^2$ V$^{-1}$ s$^{-1}$ and bandgap of 1.4 eV for the as-obtained SnS monolayers. The PFM assessment shows a very large value of ~26.1 ± 0.3 pmV$^{-1}$ for piezoelectric coefficient, which is larger than any previously reported planar structure. The SnS monolayers were applied into a piezoelectric nanogenerator, which showed a large average voltage peak output of ~150 mV at 0.7% strain. The high conversion efficiency of the liquid metal produced monolayer SnS is attributed to the large $d_{11}$ piezoelectric coefficient and single crystallinity with minimal grain boundaries, providing limited generated charge recombination sites and scattering, and consequent

high carrier mobility. This work vouches for the suitability of low-frequency energy harvesting using semiconducting monolayer SnS on flexible, wearable piezoelectric nanogenerator devices. If used in parallel arrays, due to their high voltage and current, these nanogenerators can be used in future self-powered devices either installed to receive forces from commuters or as wearables.

## Methods

**Synthesis of SnS monolayers**. A one-step synthesis procedure based on a modified liquid metal van der Waals exfoliation technique[21,22,31] was employed for the synthesis of monolayered SnS, while the interfacial layer is being exposed to H$_2$S gas. A custom-made glove-box was purged with nitrogen and vacuumed until oxygen is reduced to ppm range. Then, 50 ppm H$_2$S gas in N$_2$ background was purged into the chamber to provide the sulphide-forming environment. This atmosphere was maintained for a minimum of 60 min before and also throughout the exfoliation step. Flow rates and time durations of all periods were calculated in accordance to the volume of the glove-box chamber. Elemental tin (99.8% Roto Metals) was placed on a glass slide and reduced to its molten state by heating it on a hotplate at the temperature of 350 °C. This temperature was required to ensure the formation of SnS phase with no stoichiometric SnS$_2$ or Sn$_2$S$_3$ phases inferences. A coppery colour sulphide layer was initially formed on the surface, which was removed through pre-conditioning using heated glass slides to expose the shiny tin melt surface. The pre-conditioning step was necessary to remove any pre-existing oxides on the as-received tin. After pre-conditioning, a heated substrate such as SiO$_2$ (300 nm)/ Si was gently touched to the surface of the droplet to exfoliate the newly formed interfacial tin sulphide layer.

**Characterisation of the SnS monolayers**. TEM, HRTEM, dark field imaging, and SAED patterns were obtained using JEOL-2100F (200 kV) on a Gatan Orius

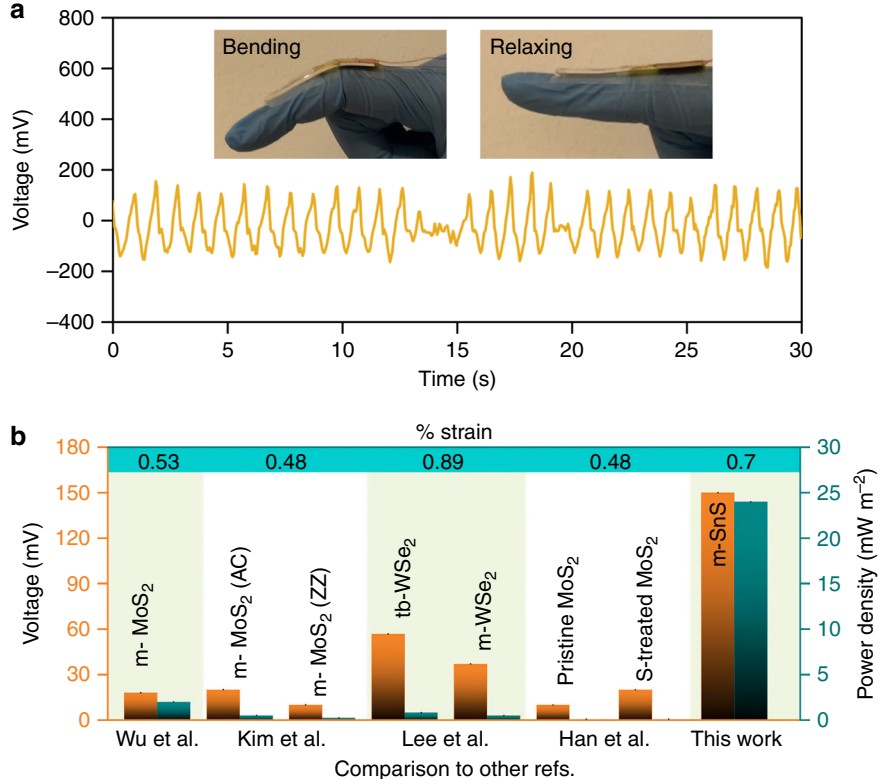

**Fig. 5 The output of the device with synthesised monolayer SnS on a flexible substrate in $d_{11}$ mode. a** Output measurement in a practical wearable device application. The voltage output of tensile bending and relaxing mode action as depicted in the inset (two-electrode device), **b** a graph showing a comparison of voltage and power density outputs of previous 2D-based piezoelectric nanogenerators[6,8,9,30] reported in comparison to our work in normalised strain (m-monolayer, S-sulphur-and tb-turbostratic).

SC1000 CCD camera and Phillips CM200, Netherland (200 kV, dark field mode with no filter). For these analyses, nanosheets were delaminated gently onto pre-heated holey Au-Carbon grids (ProScitech, Australia). Surface topography was analysed by AFM (Bruker Dimension Icon) in Scanasyst mode with a Scanasyst tip and analysed using Nanoscope software. XPS and valence band analysis were performed on a Thermo Scientific K-alpha XPS spectrometer with an Al-Kα source. Raman spectra were obtained using a Horiba Scientific LabRAM HR evolution Raman spectrometer at 532 nm laser excitation wavelength. XRD patterns were obtained using a Bruker D4 Endeavour Wide Angle XRD instrument with a wavelength of 1.54060 Å. The absorbance spectra were measured using a Cary 5000 Agilent spectroscopy system. The resulting spectra were used for computing Tauc plots to determine the optical bandgap. PESA was obtained using Riken Keiki AC-2 photoelectron spectrometer with UV light energy of 100 nW and a power law of 0.33 was used to analyse the data. The valence band maximum was established from the intersection of the linearly fitted baseline and signal slope.

The Hall mobility and carrier density of the 2D SnS at room temperature were measured using a standard technique on the fabricated SnS Hall-bar structured samples. For these, 10 nm-thick Pt electrodes were deposited by electron beam using a hard mask. The devices were then tested in a chamber consisting of a sample-in-vacuum cryostat containing a superconducting magnet within the setup. The measurements were conducted at 300 K, while the magnetic field ($B$) was varied ±2 T. The Hall carrier mobility ($\mu_H$) is $R_H \sigma_{xx}$ where $\sigma_{xx}$ is the longitudinal conductivity and the Hall coefficient $R_H = R_{xy}/B$. The hole mobility was calculated by $R_H = 1/p_e$, where $p_e$ is the hole carrier density. HR-SEM imaging was performed via FEI Verios model 460 L using through the lens low energy detector operating at 950 V.

**Piezoresponse force microscopy**. The piezoelectric measurements were performed on MFP-Infinity (Oxford Instrument, Asylum Research, Santa Barbara, CA, USA) using the lateral piezoresponse force microscopy (LPFM) mode. The substrate was pasted onto a metal chuck with Ag paste to earth the substrate and remove any surface charge that may interfere with the measurements. A conductive tip (Bruker, SCM-PIT-V2) of spring constant 3 Nm$^{-1}$ was utilised to reduce any contributions due to the effect of electrostatic discharge. The measurements were made by varying drive AC voltage amplitudes and a drive frequency of 370 kHz for the PFM mode. A background measurement was obtained to ensure the frequency had a low background contribution. The piezoelectric coefficient $d_{eff}$ was calculated by first measuring Asylum Research Periodically Poled lithium niobate (AR-PPLN) test sample with a known coefficient that can be used to calibrate the SnS since

lateral mode is measured in arbitrary units. KPFM was conducted prior to, and following, PFM imaging using the same cantilever (Electrolever) operating in AC mode and NAP mode (at a distance of 50 nm from the surface). The surface was electrically grounded to the AFM stage. In this method of imaging, the surface is first scanned with the cantilever in contact with the substrate (AC mode) and then a second pass of the same scan line is conducted at a distance of 50 nm from the surface. Here, the second pass of the surface produces a surface potential image of the surface. This provides a measure of the surface potential of the scanned area (the SnS flake and the substrate).

**Fabrication of the piezoelectric nanogenerator**. The SnS monolayers were delaminated onto either smooth fluorphlogopite mica sheets (Taiyuan Fluorphlogopite Co. Ltd., Changchun City, China) of 50 mm × 20 mm dimensions or PDMS. Reducing the thickness of the mica substrate to 20 μm ensured the flexibility of the devices. The next step involved the deposition (E-beam evaporator deposition-PVD75- Kurt J. Lesker) of Cr-Au (10/100 nm) electrodes[32]. Both two-electrode and multiple-electrode devices were fabricated. The two-electrode device is shown in Fig. 4. The electrodes have the width of 2 mm and a spacing of 40 μm separates them. Multi electrode devices consists of 15 pairs of 100 μm finger thickness and gap using a mask. The deposition hard mask was aligned to lie above the nanosheets identified under an optical microscope. Electrical wires were connected onto the electrode contact pads with silver paste. Finally, an encapsulation layer of PDMS was applied to the top of the device to passivate the surface of the material and to support the thin mica sheets. For developing the flexible nanogenerators, the same process was applied to a PDMS substrate with an optional encapsulation layer of polyamide.

**Electrical output measurements**. An oscilloscope (Agilent Technologies, DSO-X 3024A) was used for measuring the voltage variation with time across the device. A known variable resistor in parallel with the device is used to calculate the current and power outputs. Current data was measured using a KEYSIGHT B2912A precision source meter capable of measuring in 10 fA range. Data was acquired every 0.1 s using a Labview program connected to the system in an electromagnetic shielding chamber. When testing the device, it was connected to an external resistance of 1 GΩ. The same strain of ~0.7% was applied using an automated probe.

A commercial loading set-up (INSTRON Electropuls E3000 All-Electric Dynamic Test Instrument) was applied to provide a controlled and recordable mechanical compression for the flexible PDMS device and a tapping force on the

PDMS-encapsulated mica device (as represented in Supplementary Fig. 17). The impactor head, that was used to provide a tapping action, was homed to the surface of the device as the initial setpoint (i.e $d_{min} = 0$ mm) with a low pre-load of ~3 N. A square pulse of set amplitude ($\Delta d$) of 2 mm and frequency of a desired number of cycles was then applied to the device surface where displacement of the head is adjusted to $d_{max}$ of ~2 mm. The tests were done at an R-ratio of 0.43 where the minimum and maximum load was measured to be ~3 N and ~7 N, respectively. The maximum contact load and maximum displacement were recorded via data acquisition software.

## Data availability

The data that support the findings of this study are available from the corresponding author upon reasonable request.

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

## Acknowledgements

We would like to acknowledge the facilities as well as scientific and technical assistance from the staff of the Australian Microscopy & Microanalysis Research Facility (RMMF) and the Micro Nano Research Facility (MNRF) at RMIT University. This project was facilitated by funding from the Australian Research Council (ARC) Discovery Project for the study of 2D Piezoelectric materials (DP180102752). N. M. acknowledges ARC for funding received under the ARC Discovery Project scheme (DP170102138) and Vice-Chancellor fellowship scheme at RMIT University. K.K.-Z. and T.D. acknowledge the ARC Centre of Excellence FLEET (CE170100039). T.D acknowledges the funding received via the ARC DECRA scheme (DE190100100). S.P.R. acknowledges the support of the ARC Centre of Excellence in Exciton Science (CE170100026) and ARC Centre of Excellence FLEET (CE170100039). K.K.-Z. also thanks support from ARC Laureate Fellowship (FL180100053). We also acknowledge the CSIRO (Dr Anthony Chesman) for facilitation of the PESA measurements.

## Author contributions

H.K., K.K.-Z., T.D. and Y.L. conceptualised the project and synthesis methodologies. P.A. assisted H.K. with the synthesis. A.Z., M.B.G. and N.M. performed the microscopic analysis and imaging. J.Y. and B.Z. aided H.K. in acquiring XRD, UV-Vis and Raman spectra measurements. G.Z. and L.W. contributed to the Hall-effect measurements. A.R.R., V.K. and M.A.R. aided H.K. with the device design, fabrication and measurements. A.E. and H.K. performed the PFM measurements. All the authors contributed to the discussion and revision of the paper.

## Competing interests

The authors declare no competing interests.
