## [Peer Review File · Nature Communications]

Reviewers' comments:

Reviewer #1 (Remarks to the Author):

In this paper, the authors concoct a methodology to fabricate large-area monolayers of SnS---a prototypical group IV monochalcogenides. As motivated by the authors, several systems in this class of materials are piezoelectric and so is the monolayer made by the authors. The material is then characterized as is the usual practice.

I have no qualms about this paper being published in some suitable materials journal. I can see that the work will add a useful set of data to the literature. However, there is nothing in this work that prompts me to believe it belongs to a journal like Nature Communications. I have learned nothing new. There is no new invention nor basic scientific discovery. No novel physical insights. Simply a nice use of existing methods to come up with yet another approach to create large scale monolayers.

This paper should be perfectly fine in a normal materials journal which is where it belongs.

Reviewer #2 (Remarks to the Author):

In this communication, Prof. Kalantar-Zadeh and colleagues utilized a liquid metal-based synthesis method for growing monolayer SnS film and evaluated its performance as piezoelectric nanogenerator. This liquid metal-based synthesis method had been widely adopted by many research groups including Prof. Kalantar-Zadeh. In their last study, they synthesized SnO film (ACS Nano 2017, 11, 10974–10983). It is great to see this method is quite universal for growing low melting point metal-based compounds and potentially extends their applications in large-area functional devices. Here I have concerns about the size, uniformity, grain size, grain boundary, cracks, wrinkles, overlap layers and byproducts in the as-grown SnS film. However, these important data are missing in this work. Furthermore, any improvements in the abovementioned aspects of the as-grown SnS film when comparing with previous SnO film (ACS Nano 2017, 11, 10974–10983)? In addition, some figures have poor quality, I suggest the authors to replace them with high quality figures. There are also many typos in the manuscript, reference and supporting information. I would like to see all the above and following comments are well addressed before recommending publication in Nature Communications.

The following are some detailed comments:

1. What is size of the typical SnS film grown by liquid-metal based synthesis method? Please demonstrate the detail data in the revised manuscript.
2. What is the surface coverage of the transferred SnS film on target substrates? It can not be 100% when we transfer a curved film from liquid droplet to a flat substrate. How do the discontinuities (e.g., cracks, wrinkles, overlap layers and byproducts) affect the electrical (hall mobility) and piezoelectricity?
3. P6, line 109. "The near single crystallinity and low concentration of grain boundaries across the planes are also identified using dark field TEM". The TEM gives very local information of a very small area. It cannot be used to claim that the SnS film is single crystalline. What is mean grain size of SnS film, any direct data?
3. How to control the products to be SnS, instead of SnS₂, what is the key factor (e.g., growth temperature, H₂S)? If possible, I suggest the author to plot a growth map to link the products and growth conditions.
4. What is the sample for Hall mobility measurements, monolayer or few-layer? What is the dimension of the devices? Please provide a SEM image in the SI for evaluation.
5. What is the atomic ratio of Sn and S in the as-grown SnS film? Refer to the XPS data.
6. P8, line156. "The homogeneity and reduced grain boundary..." No clear data can support this claim.

7. Fig. S1. I suggest the author to prepare a schematic diagram and label all the main components of the setup. Otherwise, the readers may have difficulty to understand the experiment.

8. Fig. S2. a. Nothing can be seen in Figure S2a. I suggest the authors carefully choose the silicon substrates with proper SiO₂ thickness which can give the best contrast of monolayer and few-layer SnS film. Alternatively, I suggest the authors to use SEM image instead of optical image. b. Here, the optical images only give the information of a very small area, we cannot conclude the growth method can provide high uniformity wafer-scale SnS film without grain boundaries, cracks, wrinkles and overlap layers.

10. Fig.3. a. What are the thick particles on SnS film. Is it possible to get rid of these byproducts? b. Please label the chemical formula of the thin film and the particles in Fig. S3a.

Reviewer #3 (Remarks to the Author):

The manuscript reports the fabrication of large-area high quality monolayer SnS and their subsequent deployment in a piezoelectric generator. The SnS monolayers are fabricated using a liquid-metal based synthesis method, that allows for the production of single crystal, grain-boundary free 2D flakes of SnS, which can then be transferred to a different substrate to be made into a piezoelectric generator.

I find the fabrication method quite compelling, particularly the fact the yield of this 2D material is greatly improved. The transport measurements seem robust and point to a narrow bandgap p type semiconductor with high mobility. However, I have concerns related to the piezoelectric properties being reported, particularly the device performance and subsequent analysis. There are also other issues that I will highlight below, which I hope the authors can address.

1. In Figure 1a, the middle panel says "printed SnS" and "SnS prints" are also referred to on Page 7. It is not clear what "print" here refers to, as no printing method has been discussed in the methods section, nor in the supporting info.

2. The dimensions of the piezoelectric device are not given, and hence it is difficult to determine whether the whole device area is made up of monolayer SnS. AFM cannot possibly be used to determine the thickness over millimeter-scale samples, for example, and so Fig 2b cannot be used as proof that the whole device is indeed monolayer SnS.

3. Following the above point, there is no indication as to how good the coverage of the substrate is after the transfer process. Are the SnS monolayers connected, or are they not packed densely enough. What configuration do they take after transfer. There is no reason to believe that the atoms align as perfectly as shown in Fig 1 across the whole sample, as there does not seem to be anything in the transfer process that would guarantee the flakes remain aligned, or indeed in monolayer form. The authors should indicate how this is achieved in practice, and more importantly, how this is verified.

4. Unfortunately I am not convinced by the arguments put forward regarding the piezoelectric output. For one, the very nature of the interdigitated electrodes would negate any piezoelectric voltage developed, unless the portions of the flakes between each pair of electrodes were physically aligned opposite to its neighbour such that the dipoles were oppositely aligned. In piezoelectrics that are ferroelectric, this is achieved through electrical poling, which is not the case in the present paper. The question is, how does the piezoelectric charge developed here add up across interdigitated electrodes, without any polar direction being explicitly defined between each pair of electrodes.

5. The generated voltages due to impacting may be due to the triboelectric effect, which has not been explicitly ruled out, while that due to lateral stretching may be attributed to the flexoelectric effect. Both triboelectric and flexoelectric effects would be expected to manifest strongly in a 2D

material.

6. The authors have not actually shown any direct measurement of piezoelectric properties of their monolayers. Given that they have access to an AFM, the authors should provide piezo-response force microscopy (PFM) data to verify the piezoelectric properties, including showing how the dipoles are aligned in this material in the device configuration. This is crucial data to show the nanoscale piezoelectricity in this material. Without this, points 4 and 5 above cast doubt on the interpretation of the device data.

7. In general, comparing piezoelectric "output" across different materials or devices in the literature is meaningless without proper context. For example, what was the mechanical excitation in the other studies? The output should at least be normalised to the input mechanical stimulus to provide a fair comparison.

8. The authors have not commented at all on the frequency dependence of the electrical output that they observe in Fig 4b. More generally, the output profile shown in Fig4a(ii) would not be expected from a piezoelectric that is semiconducting, precisely due to the piezotronic effect that the authors demonstrate. One would expect carrier depletion due to the Schottky junctions to give rise to a steady-state piezoelectric response, rather than the transient response observed. This again points to the origin of the device output being something other than pure piezoelectric.

In summary, while the fabrication method is quite interesting, I find that the paper does not provide compelling evidence of piezoelectricity in this material, particularly without PFM data (e.g. it would be interesting to report the measured d coefficient). Also the device geometry and output characteristics are not consistent with piezoelectric behaviour, particularly from a semiconductor. In light of the above, I am unable to recommend acceptance of this work.

Response to the reviewers' comments

The following is a point-by-point response to the reviewers' comments.

Reviewer 1

Comments:

In this paper, the authors concoct a methodology to fabricate large-area monolayers of SnS---a prototypical group IV monochalcogenides. As motivated by the authors, several systems in this class of materials are piezoelectric and so is the monolayer made by the authors. The material is then characterized as is the usual practice.

I have no qualms about this paper being published in some suitable materials journal. I can see that the work will add a useful set of data to the literature. However, there is nothing in this work that prompts me to believe it belongs to a journal like Nature Communications. I have learned nothing new. There is no new invention nor basic scientific discovery. No novel physical insights. Simply a nice use of existing methods to come up with yet another approach to create large scale monolayers.

This paper should be perfectly fine in a normal materials journal which is where it belongs.

Our response

We understand this comment and have significantly modified the paper to better illustrate the very important novelties of this work.

This paper contains several developments that are world's first. Something that has never been demonstrated before is giant piezoelectricity in any planar crystal by an actual measurement. In fact, such a demonstration is of extraordinary value with remarkable scientific and technological outcomes. This giant piezoelectricity has been predicted in selected monolayer metal chalcogenides such as SnS due to their puckered crystal structures, but no research group has ever been able to make them as the exfoliation of these crystals has so far been impossible. Here we used a bottom up approach based on liquid metal reaction media that eventually allowed the

synthesis of monolayer SnS. We endeavored to address your comment in this version of the paper and strongly highlight the most important novelties in the introduction. In addition, we added extra measurements on assessing the piezoelectric coupling coefficient using PFM and also two electrode devices. The measured effective piezoelectric coefficient is $> 35 \text{ pmV}^{-1}$ a number that is much larger than any previously demonstrated planar crystal.

Reviewer 2

Comments:

In this communication, Prof. Kalantar-Zadeh and colleagues utilized a liquid metal-based synthesis method for growing monolayer SnS film and evaluated its performance as piezoelectric nanogenerator. This liquid metal-based synthesis method had been widely adopted by many research groups including Prof. Kalantar-Zadeh. In their last study, they synthesized SnO film (ACS Nano 2017, 11, 10974–10983). It is great to see this method is quite universal for growing low melting point metal-based compounds and potentially extends their applications in large-area functional devices. Here I have concerns about the size, uniformity, grain size, grain boundary, cracks, wrinkles, overlap layers and byproducts in the as-grown SnS film. However, these important data are missing in this work. Furthermore, any improvements in the abovementioned aspects of the as-grown SnS film when comparing with previous SnO film (ACS Nano 2017, 11, 10974–10983)? In addition, some figures have poor quality, I suggest the authors to replace them with high quality figures. There are also many typos in the manuscript, reference and supporting information. I would like to see all the above and following comments are well addressed before recommending publication in Nature Communications.

Our response

Thank you for your insightful comments. We carefully addressed your concerns on size, uniformity, grain size, grain boundary, cracks, wrinkles, overlap layers and byproducts in the as-grown monolayer SnS film by answering your detailed comments below. In terms, of your query on the improvements on the previous work on SnO¹, the work presented here is an absolutely different concept. By changing the ambient gas of the experiment, we have demonstrated that a whole new class of monolayer tin sulphide, with giant piezoelectricity, compounds can be synthesised using our method for low-melting point metals. This is particularly advantageous for materials with the prediction of giant piezoelectricity, such as SnS, which was challenging previously. This work reports an unprecedented synthesis technique in sulphur containing ambient (H₂S) that that is grown on the liquid metal interface.

This liquid metal-based method allows us to extract centimetre sized homogenous, highly crystalline monolayer SnS that finally show there is an answer to the quest for giant planar piezoelectricity in puckered structures. Previous reports for this material using the conventional

methods such as micromechanical cleavage or liquid solvent synthesis methods have not been successful as they counter trade-off between lateral dimensions and tenacity due to the large interlayer coupling. The introduction was carefully edited to highlight the novelty.

Thank you for the feedback - we have carefully edited the paper according to your comment, enhanced the quality of the figures and thoroughly checked the manuscript for typos as per your advice.

The following are some detailed comments:

- 1. What is size of the typical SnS film grown by liquid-metal based synthesis method?
Please demonstrate the detail data in the revised manuscript.**

Our response

We appreciate this comment as addressing it enhances the quality of this report. The synthesis method applied in this paper involved the mechanical transfer of a sulphide layer formed on the surface of the liquid metal droplet. As the synthesis process follows the self-limiting Cabrera-Mot model², this sulphide layer is evenly formed over the whole surface of the droplet and hence the size of sheet delaminated is proportional to the surface area of the droplet and can be controlled as such in this manner.

We added the following text to our manuscript to help readers envision the scale of the SnS film produced. We have also added some more optical images in the Supplementary Information to further illustrate the large size synthesised product.

Added text in the main manuscript:

“While the size of the monolayer SnS can be potentially very large, in this work, we consistently used an average droplet size of 1 cm in diameter and a 45° angle of contact resulting in monolayer sheets of hundreds of microns in dimensions. The mechanical exfoliation parameters were used diligently in to obtain consistent results and were the most suitable parameters for the custom-designed setup we used for the synthesis (taking into account the size of the heater and workspace area in the glovebox) but are easily scalable to any setup.”

Added figure in the Supplementary Information:

Fig S2. Optical images of the monolayer SnS synthesised by the delamination of sulphide layer showing no cracks and consistent colour across the films, evidencing that the films thickness do not change and remain monolayer in micron size dimensions. Scale bar is 10 μm .

2. What is the surface coverage of the transferred SnS film on target substrates? It can not be 100% when we transfer a curved film from liquid droplet to a flat substrate. How do the discontinuities (e.g., cracks, wrinkles, overlap layers and byproducts) affect the electrical (hall mobility) and piezoelectricity?

Our response

Thank you for your comment and we agree that since the transfer is from a curved surface to a flat one, the transfer is not 100%. After many tests, we realized that the actual % transfer is dependent on the angle of contact of the substrate with the liquid metal droplet surface and also the substrate to droplet size ratio (surface area of contact region).

The factors affecting transfer % have been illustrated in **Fig. S3** below and a detailed explanation added to the Supplementary Information as **Supplementary Note S1**. to make it clearer to the readers.

Supplementary Note S1. Factors affecting percentage transfer of SnS sheet to the substrate.

“In the exfoliation/transfer process, the actual % transfer depends on the angle of contact of the substrate with reference to the liquid metal droplet surface and the substrate to droplet size ratio (surface area of contact region).

These parameters taken into account, as well as the force with which the substrate makes contact is the basis for the optimisation of the synthesis process. Our optimisation aimed towards the synthesis of large area, homogenous monolayer sheets. Cracks could arise in the use of excessive force to contact the surface while wrinkles were mostly limited to the edges of the sheet due to the folding of the sheet during the transfer. Following such parameters, the transferred area consisted of clean, homogenous monolayer sheets of SnS and the measurements were made focused on these areas. The synthesis method was further optimised to ensure no formation of byproducts by controlling the temperature and transfer time and hence these did not affect the process.”

The following diagram was also added to the Supplementary Information as **Fig. S3**.

Fig. S3 Schematic illustration of factors affecting percentage transfer of SnS sheet to the substrate.

To also, further explain the optimised synthesis, a product growth map has been added to the Supplementary information with reference to comments #4 and #9.

3. P6, line 109. “The near single crystallinity and low concentration of grain boundaries across the planes are also identified using dark field TEM”. The TEM gives very local information of a very small area. It cannot be used to claim that the SnS film is single crystalline. What is mean grain size of SnS film, any direct data?

Our response

Thank you for your comment. To address your comment, we endeavoured to take SAEDs across various locations on the synthesised sheets to show the consistent crystallographic orientations suggesting the single crystalline characteristic as other less localized methods are unavailable to us. From our analysis, (which is across sheets of several microns due to the limitations of transfer onto TEM grids), we have consistently seen single crystalline virtues. However, we cannot give any direct data for the grain size of a single SnS film from this except for saying they are larger than several microns for sure.

We agree that the suggested statement may be misleading and have changed it to the following:

Main manuscript:

“The near single crystallinity, at least within micron domains, and low concentration of grain boundaries across the planes are also identified using dark field transmission electron microscopy (TEM) (**Fig. 1d**). We conducted SAED measurements across various locations on the synthesised/transferred monolayers of SnS which confirmed consistent crystallographic orientations, suggesting the single crystalline characteristic (**Supplementary Note S2 and Fig. S5**).”

Supplementary Information:

Supplementary Note S2: Further analysis of single-crystallinity, reduced grain boundaries and homogeneity of as-synthesised SnS sheets.

Selected-area electron diffraction (SAED) patterns at 4 typical locations on the transferred sheet (marked 1 to 4 in **Fig S5 a**) were generated to assess the single crystallinity of these monolayers. These areas showed the same crystallographic orientations (**Fig. S5 b-e**) and the corresponding lattice fringe patterns (**Fig. S5 f-i**) taken zoomed into these areas. These further characterizations

indicate the large area homogenous single crystalline domains (at least $>1 \mu\text{m}$) and minimal grain boundaries in the sheets. The two adjacent sheets chosen for this analysis were deliberately chosen to demonstrate that the sheets from one touch of the grid result in large area single crystals even if the sheets break during the synthesis process. The transfer from the droplet to the amorphous carbon filmed grid is not as ideal as the transfer onto a SiO_2/Si substrate hence the sheets on the TEM grid are broken and comparatively smaller than the ones obtained on rigid substrates.

Fig. S5 a) TEM image of a large area SnS nanosheet transferred onto a TEM grid. b)–e) Four typical SAED patterns collected from the areas labeled 1–4 in a presenting closely matched lattice orientations over the whole sheet while f)–i) shows the corresponding lattice fringe spacings when zoomed into indicated areas on the sheet.

Please also refer to the response to your comment #7.

4. How to control the products to be SnS, instead of SnS₂, what is the key factor (e.g., growth temperature, H₂S)? If possible, I suggest the author to plot a growth map to link the products and growth conditions.

Our response

Thank you for this important comment. We have drawn the following growth map that links the products and growth conditions of the SnS and SnS₂. This has been added to the Supplementary figure for the readers in order to further explain the optimisation of the synthesis process.

Fig. S4. Growth map illustrating the link between growth conditions and products of liquid metal based -sulphide synthesis.

Also, the following was added to the manuscript:

“A growth map illustrating the link between growth conditions and obtaining SnS or SnS₂ products is presented in Supplementary Information (**Fig. S4**).”

5. What is the sample for Hall mobility measurements, monolayer or few-layer? What is the dimension of the devices? Please provide a SEM image in the SI for evaluation.

Our response

Thank you for this important comment. The Hall mobility measurements are for the monolayer sample. The dimensions of the fabricated device have been illustrated in the Supplementary figure below including the optical image of the device used for the measurement. (SEM was unable to cover the whole device so an optical image is provided instead).

Fig. S10 Hall-effect mobility measurement: a) Optical image of the device with scale bar of 2 mm and zoom in of electrodes with SnS monolayer and b) electrode configuration with dimensions used. The image of the actual electrodes were taken at an angle therefore look slanted but are in fact like the schematic.

6. What is the atomic ratio of Sn and S in the as-grown SnS film? Refer to the XPS data.

Our response

Thank you for your comment. Due to reasons we will discuss, the accurate extraction of the ratio of the Sn: S is challenging for monolayer samples using XPS but we used TEM-EDS to do so instead. The following was added to the Supplementary Information as **Supplementary Note S3**:

Supplementary Note S3: Discussion on calculation of Sn: S ratios

“We found that the ratio obtained by XPS always shows that S is less than Sn. The loss of sulphur may be due the intensity of the X-ray resulting in a depleted S ratio due to the extremely thin layer as this is also observed in self-assembling monolayers. Another factor is the overlapping of the Si2s plasmon peak with the S2p spectra. Mostly, interpreting the S2p spectrum on silicon-based substrates is challenging due to the satellite peaks induced from the proximal Si2s (~152 eV), which is a result of surface plasmon (~17 eV) being excited by the photoelectrons³⁻⁶. Hence, obtaining a clean S2p peak is often difficult on silicon surfaces which distorts ratio analysis to some extent. The lower S content as well as the correct binding values of the Sn for the 2+ valency state confirms that the synthesised material is not SnS₂ which is also confirmed by Raman. We have further conducted EDS mapping to calculate the ratio instead (**Fig. S9**).”

Considering the abovementioned issue, EDS was conducted as a complementary assessment for realizing the Sn and S ratios. The ratio was very close to 1:1 as can be seen in the figure below that was included in the Supplementary Information:

Fig. S9 EDS of monolayer SnS- Elemental mapping of a) Sn b) S with c) spectrum that shows near 1:1 ratio by spectrum. Scalebar is 100nm

7. P8, line156. “The homogeneity and reduced grain boundary...” No clear data can support this claim.

Our response

Thank you for your comment. We agree that the current data does not necessarily verify this. We have thus expanded our TEM analysis to more points across the sheet and on various sheets to demonstrate this claim using a similar method to previous reports⁷⁻⁹. TEM grids are less ideal substrates for a direct mechanical exfoliation of the 2D sheets in comparison to that of oxygen-terminated substrates such as SiO₂. This is due to the reduced van der Waals adhesion forces, existence of holes and the presence of the copper support. However, several micrometer monolayer sheets can be successfully transferred onto the holey carbon TEM grids featuring single crystallinity. During the transfer, larger sheets tend to break into smaller pieces, however as shown in the figure, all of the segments belong to a single crystal by several SAED measurements across different regions of the sheets.

Please refer to comment #3 for the extra discussions and measurements.

8. Fig. S1. I suggest the author to prepare a schematic diagram and label all the main components of the setup. Otherwise, the readers may have difficulty to understand the experiment.

Our response

Thank you for your suggestion. We have drawn the following schematic and have annotated it with the main components of the setup in order to enhance the reader's understanding of the synthesis experiment. The following image was added to the Supplementary Information.

Fig. S1 The schematic of the custom-made set-up to provide an ambient S-saturated environment (using H_2S in ambient N_2) for the sulphide layer formation and its delamination.

9. Fig. S2. a. Nothing can be seen in Figure S2a. I suggest the authors carefully choose the silicon substrates with proper SiO_2 thickness which can give the best contrast of monolayer and few-layer SnS film. Alternatively, I suggest the authors to use SEM image instead of optical image. b. Here, the optical images only give the information of a very small area, we cannot conclude the growth method can provide high uniformity wafer-scale SnS film without grain boundaries, cracks, wrinkles and overlap layers.

Our response

Thank you for the comment. We appreciate that the image previously inserted was not of good quality to illustrate the contrast of the 2D SnS on the SiO_2/Si substrate. We have added an extra figure with a few optical images showing SnS in contrast to the SiO_2/Si substrate. b. We have tried to add some figures with extra-large sheets as well. Overlap layers are avoided by synthesising at the optimised conditions whereas cracks and wrinkles occur in a very low percentage of samples. Wrinkles and folds are restricted to the edges and occur mainly on TEM grids. TEM grids are less ideal substrates for the direct mechanical exfoliation of 2D sheets than that of oxygen terminated substrates such as SiO_2 due to the lower van der Waals attractive forces, existence of holes and copper support.

Fig. S2 Optical images of the monolayer SnS synthesised by delamination of sulphide layer showing no cracks and consistent colour across the films, evidencing that the films thickness do not change and remain monolayer in micron size dimensions. Scale bar is 10 μm .

10. Fig.3. a. What are the thick particles on SnS film. Is it possible to get rid of these byproducts? b. Please label the chemical formula of the thin film and the particles in Fig. S3a.

Thank you for these comments. The thick particles are islands of SnS₂ on a thin sheet of SnS. These are produced on the providing excessive heat or prolonged exposure to H₂S. In our experimental setup, because the concentration of the gas is maintained, these islands are produced when the heater temperature reaches values above 375°C or when the sulphurised layer on the droplet is not removed within 6 seconds. Double layers of sulphurised tin start to grow and hence there is a mix of SnS and SnS₂. We have not focused on this investigation in this work as we optimised the synthesis conditions to produce pure thin SnS sheets as required for our application. This further study, in detail may be possibly be an avenue to explore and publish as a follow on to this work at a later stage. But from our optimisation process, we are happy to illustrate the parameters in the growth map diagram below and have added this to the Supplementary Information.

b. **Fig. S4** (previously **Fig. S3**) and its caption (revised figure shown below) has been further annotated to include the chemical formula of the film and particles as per suggested by the reviewer and incorporated into the growth map below.

Fig. S4 Growth map illustrating the link between growth conditions and products of liquid metal based -sulphide synthesis. Inset into each region is TEM image and SAED pattern. The SnS₂ can be indexed to hexagonal Berndtite-4H SnS₂ (PDF card no: 21-1231). This is a similar to the emergence of stoichiometric SnO₂ islands on SnO sheets that has been reported when exposing liquid tin to oxygen for a long time^{1,10}.

Reviewer 3

Comments:

The manuscript reports the fabrication of large-area high quality monolayer SnS and their subsequent deployment in a piezoelectric generator. The SnS monolayers are fabricated using a liquid-metal based synthesis method, that allows for the production of single crystal, grain-boundary free 2D flakes of SnS, which can then be transferred to a different substrate to be made into a piezoelectric generator.

I find the fabrication method quite compelling, particularly the fact the yield of this 2D material is greatly improved. The transport measurements seem robust and point to a narrow bandgap p type semiconductor with high mobility. However, I have concerns related to the piezoelectric properties being reported, particularly the device performance and subsequent analysis. There are also other issues that I will highlight below, which I hope the authors can address.

Our response

Thank you for your kind words and important comments on the technical parts in our paper regarding the assessment of piezoelectric properties. We read your comments and carefully addressed them.

- 1. In Figure 1a, the middle panel says "printed SnS" and "SnS prints" are also referred to on Page 7. It is not clear what "print" here refers to, as no printing method has been discussed in the methods section, nor in the supporting info.**

Our response

Thank you for the comment – we can see why this may have been confusing and have added the relevant description to the manuscript text. We used printing as the process for transferring the SnS monolayers to the substrate by touching the surface of the exposed liquid metal droplet and printing onto it. To address your comment, we have changed the word ‘print’ to ‘transferred’ or ‘mechanical transfer’ where relevant in the manuscript to rectify this. **Fig. 1** has been updated as well as below.

Fig. 1. Synthesis schematic representation and material characterisations of the 2D SnS: a) schematical depiction of the synthesis process, monolayer of SnS and its application on a nanogenerator transducer. b) TEM image of monolayered SnS nanosheet c) HRTEM fringe pattern showing d -spacing of 0.29 nm matched to the plane (101) with inset that shows SAED pattern also indicating a 0.29 nm d -spacing matched to the (101) plane d) dark field TEM image of a 2D SnS that shows very low presence of grain boundaries, confirming near single crystallinity of the planes.

The line previously containing print has been changed from “X-Ray diffraction (XRD) of multiple SnS prints re-confirms the growth of orthorhombic SnS, matching PDF card No. 39-0354 (Fig. S4).” to “X-Ray diffraction (XRD) of multiple SnS transfers re-confirms the growth of orthorhombic SnS, matching PDF card No. 39-0354 (Fig. S6).”

2. **The dimensions of the piezoelectric device are not given, and hence it is difficult to determine whether the whole device area is made up of monolayer SnS. AFM cannot possibly used to determine the thickness over millimeter-scale samples, for example, and so Fig 2b cannot be used as prove that the whole device is indeed monolayer SnS.**

Our response

Thank you for this important comment. We have now added the following text to our Methods section to clearly discuss the dimensions of the piezoelectric device.

Main text:

“The next step involved the deposition (E-beam evaporator deposition-PVD75- Kurt J. Lesker) of Cr-Au (10/100 nm) interdigital electrodes with 15 pairs of 100 μm finger thickness and gap using a mask. For the two-electrode device, a similar process was followed using a mask with a 40 μm gap. The deposition hard mask was aligned to lie above the nanosheets identified under an optical microscope. Electrical wires were connected onto the electrode contact pads with silver paste.”

We agree that the since AFM is limited to approximately 100 μm scan sizes at a time, this method is not ideal to determine the thickness of the millimeter-scale sheet for the whole device. We have obtained a large number of AFM images that all show consistent monolayer (not shown for brevity). In order to address your comment with another characterisation technique we added optical images of the large flakes. The homogeneity of the monolayer sheets (all have the same color and contrast) under the microscope warrants this conclusion as well the fact that the synthesis conditions are highly optimised for the monolayer synthesis.

Fig. S2 Optical images of the monolayer SnS synthesised by delamination of sulphide layer showing no cracks and consistent colour across the films, evidencing that the films thickness do not change and remain monolayer in micron size dimensions. Scale bar is 10 μm .

- 3. Following the above point, there is no indication as to how good the coverage of the substrate is after the transfer process. Are the SnS monolayer connected, or are they not packed densely enough. What configuration do they take after transfer. There is no reason to believe that the atoms align as perfectly as shown in Fig 1 across the whole sample, as there does not seem to be anything in the transfer process that would**

guarantee the flakes remain aligned, or indeed in monolayer form. The authors should indicate how this is achieved in practice, and more importantly, how this is verified.

Our response

Thank you for your suggestion. As per our discussions in response to your comment #2, we added optical images of the flakes at different locations that show homogenous sheets with no cracks and wrinkles. A large number of AFM imaging on the edge of the sheets also consistently show monolayers as well (not shown for brevity).

4. Unfortunately I am not convinced by the arguments put forward regarding the piezoelectric output. For one, the very nature of the interdigitated electrodes would negate any piezoelectric voltage developed, unless the portions of the flakes between each pair of electrodes were physically aligned opposite to its neighbour such that the dipoles were oppositely aligned. In piezoelectrics that are ferroelectric, this is achieved through electrical poling, which is not the case in the present paper. The question is, how does the piezoelectric charge developed here add up across interdigitated electrodes, without any polar direction being explicitly defined between each pair of electrodes.

Our response

Thank you for your this very important comment. We agree that using interdigitated electrodes is not the ideal configuration to use when measuring responses in such devices and hence based on your comment, we remade the device to a single pair electrode configuration.

We replaced this figure into the manuscript and based on the obtained voltages adjusted the tables and discussions.

Main text:

Fig. 5. The outputs of the devices with synthesised monolayer SnS using tapping mode to excite d_{31} : a) the optical image of the two-electrode device. b) response of the output voltage of the two-electrode device. c) response of the output voltage of the multi-electrode device excited at 5 Hz. d) A graph showing the variation of average output power and voltage at varying tapping frequencies and inset is the photo of the PDMS-encapsulated mica flexible nanogenerator (multi-electrode device). e) Stability test of output voltage with stimulus applied at frequency 5 Hz for ~4,700 cycles (multi-electrode device).

“Both two-electrode and multiple-electrode devices were fabricated. The two-electrode device is shown in **Fig. 5**. The electrodes have the width of 2 mm and the spacing of 40 μm separates them.”

Fig. 6. The output measurements of the device with synthesised monolayer SnS on a flexible substrate in d_{11} mode, for practical wearable device applications: a) voltage output of tensile bending and relaxing mode action as depicted in the inset, (two-electrode device) b) a graph showing a comparison of voltage and power / unit length outputs of previous 2D-based piezoelectric nanogenerators¹²⁻¹⁵ reported in comparison to our work in normalised strain. **Note:** Power/unit length calculations were computed by taking the ratio of the maximum instantaneous power and the electrode length of the device.

5. The generated voltages due to impacting may be due to the triboelectric effect, which has not been explicitly ruled out, while that due to lateral stretching may be attributed to the flexoelectric effect. Both triboelectric and flexoelectric effects would be expected to manifest strongly in a 2D material.

Our response

Thank you for your suggestion, Triboelectric effect was ruled out by measuring the response of a device without the 2D material and with a 2D oxide material deposited between the electrodes (Fig. S18).

Fig. S18. Signal measured after applying mechanical stimuli with a) no monolayer SnS in between the two electrodes and b) also when SnO was place between the two electrodes.

As for flexoelectric, this phenomenon is generally not dominant in monolayer structures due to a diminished strain gradient. Also, the PFM already presented in the SI that shows minimal out of

plane piezoelectricity. This favors the possibility of piezoelectricity presence rather than flexoelectricity as the source of the response.

The following was added to the manuscript:

“Both triboelectric and flexoelectric effects can be ruled out for the generation of the outputs. Triboelectric effect is ruled out after showing that devices with no materials between electrodes or devices with SnO between electrodes could not generate any voltage after applying mechanical stimuli (**Fig. S18**). As for flexoelectric, this phenomenon is diminished in monolayer structures due to no strain gradient. (**Fig. S19**)”

The following figure was also added to the Supplementary Information (**Fig. S19**):

Fig. S19 PFM measurement of SnS showing a) negligible out-of-plane response in Vertical PFM mode and comparatively higher in-plane response in b) Lateral PFM mode. This is expected due to the structure of the crystal. Any in-plane effect is exhibited due to piezoelectric effect whereas the negligible out-of-plane effect might be attributed to flexoelectricity¹¹ but this phenomenon is not dominant in monolayer structures due to the diminished strain gradient and hence not considered a dominant contributor in the device measurements.¹⁶ Also, any such effects are more prominent in free-standing measurements. Scalebar is 8 μ m.

6. The authors have not actually shown any direct measurement of piezoelectric properties of their monolayers. Given that they have access to an AFM, the authors should provide piezo-response force microscopy (PFM) data to verify the piezoelectric properties, including showing how the dipoles are aligned in this material in the device configuration. This is crucial data to show the nanoscale piezoelectricity in this material. Without this, points 4 and 5 above cast doubt on the interpretation of the device data.

Our response

Thank you for your suggestion, you are correct and such measurements should have been shown in the first version of the manuscript. Fortunately, the PFM unit of our system is now operational, and we could verify the piezoelectricity data.

The following was added to the manuscript:

Fig. 4 Lateral piezoelectric response of the monolayer SnS a) Topography of the monolayer sheet with the height profile of the sheet illustrating it to be ~1 nm, which is approximately a monolayer. This is also confirmed by the histogram showing the height distribution b) Lateral amplitude of the SnS monolayer with inset of histogram c) Lateral phase of the SnS monolayer with inset of histogram d) Lateral piezoresponse of monolayer SnS and PPLN vs the driving voltage. Scale bar is 1 μm .

The following was added to the manuscript Results and Discussions:

“Piezoresponse force microscopy (PFM) analysis was used for exploring the piezoelectric property of monolayer SnS as a complementary assessment. An out-of-plane coefficient is not expected due to its structure so the lateral PFM (LPFM) mode was employed to measure in the in-plane coefficients. **Fig. 4a** shows the topography of the synthesised SnS monolayer with a height profile of ~1 nm confirming the monolayer nature of the films. An inset histogram also shows the height distribution across the scan. **Fig. 4b and c** show a clear contrast between the SnS

monolayer and the SiO₂/Si substrate when measuring the lateral amplitude and phase. **Fig. 4d** shows the lateral piezoresponse as a function of the magnitude of the driving voltage that is applied to the SnS monolayer and the periodically-poled lithium niobite (PPLN) test sample. The trend shows a distinct linear increase in the piezoresponse as applied voltage increases for both samples. This confirms the source is piezoelectric and therefore the slope can be used for calculating the effective piezoelectric coefficient. The plot is fitted to a linear line which gives 0.5365 and 0.068 arb. unitsV⁻¹ for the monolayer SnS and the PPLN sample, respectively. Because $d_{\text{eff}} = 4.53 \text{ pmV}^{-1}$ for the PPLN in the lateral direction (**Fig. S12**), an effective coefficient of 35.66 pm/V is obtained for monolayer SnS. These PFM results confirm that the synthesised SnS monolayer shows a relatively strong in-plane piezoelectricity.”

Methods

Piezoresponse Force Microscopy (PFM)

The piezoelectric measurements were performed on MFP-Infinity (Oxford Instrument, Asylum Research, Santa Barbara, CA, USA) using the lateral piezoresponse force microscopy (LPFM) mode. The substrate was pasted onto a metal chuck with Ag paste to earth the substrate and remove any surface charge that may interfere with the measurements. A conductive tip (Bruker, SCM-PIT-V2) of spring constant 3 Nm⁻¹ was utilised to reduce any contributions due to the effect of electrostatic discharge. The measurements were made by varying drive AC voltage amplitudes and a drive frequency of 370 kHz for the PFM mode. A background measurement was obtained to ensure the frequency had a low background contribution. The piezoelectric coefficient d_{eff} was calculated by first measuring Asylum Research Periodically Poled lithium niobate (AR-PPLN) Test Sample with a known coefficient that can be used to calibrate the SnS since lateral mode is measured in arbitrary units.

Fig. S12 PPLN sample used for calibration a. Height/ topography image b. PFM lateral amplitude c. PFM lateral phase

Figure showing the way the measurements were made.

Fig. S13 PFM measurement mechanism for the lateral mode

7. In general, comparing piezoelectric "output" across different materials or devices in the literature is meaningless without proper context. For example, what was the mechanical excitation in the other studies? The output should at least be normalised to the input mechanical stimulus to provide a fair comparison.

Our response

Thank you for your invaluable comment. In order to provide more context to the testing conditions of each literature, we have added the % strain at which the reported maximum performance was shown to the graph. As the piezoelectric output of the devices depends on a combination of device parameters and mechanical conditions¹⁷, and these can never be the same or easily normalized to each device, we have endeavored to provide more details on the device structure and test conditions in order to evaluate the device performance as informative as possible for a meaningful comparison by adding **Table S2** to the Supplementary Information.

Table S2. Table of important parameters of previous reports on 2D material-based nanogenerators

2D Material	V	I	Maximum instantaneous power @ stated load resistance	Test conditions	Length of electrode	Power/ Unit length	Ref	Piezoelectric coefficients	
	mV	pA	pW	% strain, tapping force applied, frequency	μm	nW/m		Theoretical	Experimental
	pm/V								
Monolayer-MoS ₂	18 (peak OC voltage)	27 (peak SC current)		0.64%			23	$d_{11}=3.65$ ²⁴	
	3.5 (calc)	12.5 (calc)	55.3 fW (220 MΩ)	0.53%	25	2.2		$d_{11}=4.94$ ²⁵	
Monolayer-MoS ₂ (armchair direction)	20 (peak output)	30 (peak output)		0.5 Hz, 0.48 %, 70mm			27	$d_{11}=3.65$ ²⁴	$d_{11}=3.78$ (AC) ²⁷

Monolayer-MoS ₂ (zigzag direction)	10 (peak output)	20 peak output							$d_{11}=1.38 (ZZ)^{27}$
Monolayer-MoS ₂ (armchair direction)	20 (1 GΩ) graph		0.4 (20mV at 1 GΩ)-graph		70	5.7			
Monolayer MoS ₂ (zigzag direction)	10 mV (1 GΩ) graph		0.2 (10 mV @ 1 Gohm)-graph		50	4			
Monolayer-WSe ₂	~35 (calc)	72 (calc)	2.54 (500 MΩ)	0.39%	100	25.4	28	$d_{11}=2.79$	$d_{11}=3.26 \pm 0.3_{28}$
Turbostratic-WSe ₂	~57 (graph)	71 (calc)	4.05 @ ~0.1 GΩ	0.89%	100	40.5			$d_{11} = 0-1.5^{28}$

Monolayer-WSe ₂	~37 (graph)	55 (calc)	2.05 @ ~ 0.5 GΩ	0.89%	100	20.5			
Pristine MoS ₂	10 @1GΩ	30 (output) @1GΩ		0.48%			29	$d_{11}=3.65$ ²⁴ $d_{11}=4.94$ ²⁵ $d_{11}=3.73$ ²⁶	$d_{11}=3.06 \pm 0.6$ ²⁹
S-treated MoS ₂	20 @1GΩ	100 (output) @1GΩ		0.48%					$d_{11}=3.73 \pm 0.2$ ²⁹
Pristine MoS ₂	~ 9 (graph) but 6 mV (calc)	12 (calc)	0.07 (500 MΩ)	0.48%	100	0.7		$d_{11}=3.65$ ²⁴ $d_{11}=4.94$ ²⁵ $d_{11}=3.73$ ²⁶	$d_{11}=3.06 \pm 0.6$ ²⁹
S-treated MoS ₂	~ 27 (graph) and 19 (calc)	38.2 (calc)	0.73 (500 MΩ)	0.48%	100	7.3			$d_{11}=3.73 \pm 0.2$ ²⁹
Monolayer-SnS	150	15 nA	2.25 nW (10 MΩ)	0.70%	2 mm	112.5	This work	$d_{11}=144.76$ ³⁰ $d_{12}=-22$ ³⁰	PFM measured value of 35.66

Note: Values in green were calculated or interpreted by the authors.

AC- armchair, ZZ- zigzag

8. The authors have not commented at all on the frequency dependence of the electrical output that they observe in Fig 4b. More generally, the output profile shown in Fig4a(ii) would not be expected from a piezoelectric that is semiconducting, precisely due to the piezotronic effect that the authors demonstrate. One would expect carrier depletion due to the Schottky junctions to give rise to a steady-state piezoelectric response, rather than the transient response observed. This again points to the origin of the device output being something other than pure piezoelectric.

Our response

Thank you for your important suggestion, and we hope that the experiments that we have included with two parallel electrodes and the extra tests that rule out triboelectric effect have addressed your comment.

Also, thanks for the comment regarding the applied pressure graph. That was a graph automatically produced by the software of the equipment that we used in tapping mode. After fast imaging, we realized that the actual applied force by the tip of the equipment was not continuous and rather like shorter impulses. As such, in order to avoid any mistake, we removed the incorrect applied force graph.

To address your valuable comment on the frequency dependence of the device, we have added the following text to the manuscript:

“The device which as such showed a high output voltage would be suitable for practical energy harvesting applications which arise from randomly available mechanical stimuli. The envisioned utility of such devices is for the energy harvesting of naturally occurring excitation in the environment rather than a forced or regular driving force. We hence tested the nanogenerator in a range of frequencies. This suggests that this nanogenerator can be applied in scenarios with low frequency stimulations such as biomedical sensors, or walkways in malls or airports. It was observed that the frequency of 5 Hz gave a maximum output value of ~190 mV on average, resulting in the maximum average peak power output of 3.8 nW (not normalised by the length of electrodes). Above this, the reason the signal is damped which is most likely due to the inability of the flexible substrate to respond at the excitation rate of the stimulus force being applied and hence the output voltage is in turn reduced. However, this can be easily overcome by better choice of flexible material for the device.”

References

- 1 Daeneke, T. *et al.* Wafer-Scale Synthesis of Semiconducting SnO Monolayers from Interfacial Oxide Layers of Metallic Liquid Tin. *ACS Nano* **11**, 10974-10983 (2017).
- 2 Cabrera, N. & Mott, N. F. Theory of the oxidation of metals. *Rep. Prog. Phys.* **12**, 163-184 (1949).
- 3 Zhang, Y. F. *et al.* Electronic structure of silicon nanowires: A photoemission and x-ray absorption study. *Physical Review B* **61**, 8298-8305 (2000).
- 4 Mannella, N., Gabetta, G. & Parmigiani, F. Plasmon energy shift in porous silicon measured by x-ray photoelectron spectroscopy. *Applied Physics Letters* **79**, 4432-4434 (2001).
- 5 Kamineni, H. S. *et al.* Optical and structural characterization of thermal oxidation effects of erbium thin films deposited by electron beam on silicon. *Journal of Applied Physics* **111**, 013104 (2012).
- 6 Lee, C.-H., Chen, W.-C. & Khung, Y. XPS Analysis of 2-and 3-Aminothiophenol Grafted on Silicon (111) Hydride Surfaces. *Molecules* **23**, 2712 (2018).
- 7 Zhou, H. *et al.* Chemical vapour deposition growth of large single crystals of monolayer and bilayer graphene. *Nature Communications* **4**, 2096 (2013).
- 8 Jiang, S. *et al.* Direct synthesis and in situ characterization of monolayer parallelogrammic rhenium diselenide on gold foil. *Communications Chemistry* **1**, 17, doi:10.1038/s42004-018-0010-6 (2018).
- 9 Na, C. W., Park, S.-Y., Chung, J.-H. & Lee, J.-H. Transformation of ZnO Nanobelts into Single-Crystalline Mn₃O₄ Nanowires. *ACS Applied Materials & Interfaces* **4**, 6565-6572 (2012).
- 10 Atkin, P. *et al.* Evolution of 2D tin oxides on the surface of molten tin. *Chem. Commun.* **54**, 2102-2105 (2018).
- 11 Wang, X. *et al.* Probing Effective Out-of-Plane Piezoelectricity in van der Waals Layered Materials Induced by Flexoelectricity. *Small* **0**, 1903106.

- 12 Lee, J.-H. *et al.* Reliable Piezoelectricity in Bilayer WSe₂ for Piezoelectric Nanogenerators. *Adv. Mater.* **29**, 1606667 (2017).
- 13 Han, S. A. *et al.* Point-Defect-Passivated MoS₂ Nanosheet-Based High Performance Piezoelectric Nanogenerator. *Adv. Mater.* **30**, 1800342 (2018).
- 14 Wu, W. *et al.* Piezoelectricity of single-atomic-layer MoS₂ for energy conversion and piezotronics. *Nature* **514**, 470 (2014).
- 15 Kim, S. K. *et al.* Directional dependent piezoelectric effect in CVD grown monolayer MoS₂ for flexible piezoelectric nanogenerators. *Nano Energy* **22**, 483-489 (2016).
- 16 Zubko, P., Catalan, G. & Tagantsev, A. K. Flexoelectric Effect in Solids. *Annual Review of Materials Research* **43**, 387-421 (2013).
- 17 Chen, J. *et al.* Output characteristics of thin-film flexible piezoelectric generators: A numerical and experimental investigation. *Applied Energy* **255**, 113856 (2019).
- 18 Blonsky, M. N., Zhuang, H. L., Singh, A. K. & Hennig, R. G. Ab Initio Prediction of Piezoelectricity in Two-Dimensional Materials. *ACS Nano* **9**, 9885-9891 (2015).
- 19 Alyörük, M. M., Aierken, Y., Çakır, D., Peeters, F. M. & Sevik, C. Promising Piezoelectric Performance of Single Layer Transition-Metal Dichalcogenides and Dioxides. *The Journal of Physical Chemistry C* **119**, 23231-23237 (2015).
- 20 Duerloo, K.-A. N., Ong, M. T. & Reed, E. J. Intrinsic Piezoelectricity in Two-Dimensional Materials. *J. Phys. Chem. Lett.* **3**, 2871-2876 (2012).
- 21 Fei, R., Li, W., Li, J. & Yang, L. Giant piezoelectricity of monolayer group IV monochalcogenides: SnSe, SnS, GeSe, and GeS. *Appl. Phys. Lett.* **107**, 173104 (2015).

Reviewers' comments:

Reviewer #2 (Remarks to the Author):

The authors addressed most of the questions I asked.

One more question is about the growth map. From the growth map, can I conclude that the as-grown SnS film change to SnS₂ if the film exposes to H₂S for a time longer than ~17 min? Do this mean the as-grown SnS film is chemically unstable? I suggest the authors prepare a more accurate growth map.

The other question is still about the morphology of the SnS film. Optical images (the updated images) can only give poor contrast of the film. High-quality SEM images are required to prove the overall quality of the transferred SnS film. I suggest to authors present a group (array) of SEM images of the SnS film with a size of few tens micrometers by a few tens micrometers.

Reviewer #3 (Remarks to the Author):

The manuscript by Khan et al. reports on liquid metal-based synthesis of monolayer SnS, and their resulting piezoelectric properties. The authors have made changes based on reviewers' comments, however, there still remains significant issues. With regards the fabrication process itself, previous reviewers have raised concerns which the authors seem to have adequately addressed in their revised version. However, I am still not convinced about the validity of the measurements of piezoelectric properties, and here I feel there are major gaps in the understanding of the phenomenon. The fact that data related to the output of the interdigitated electrode geometry had to be removed without explanation raises severe concerns. It is highly unsatisfactory to simply remove data that was put forward as evidence of piezoelectricity, without explaining what the measured signals were in the first place. This, unfortunately raises doubts about subsequent measurements and interpretation. Also, the authors seem to have confused the piezoelectric and piezotronic effects, which are distinct effects in semiconductors, and there is insufficient discussion of the latter in this respect.

I have concerns regarding the PFM measurements. In contact-mode operation, there is significant electrostatic contribution which can be considerable particularly for thin 2d materials. This effect has to be corrected for by performing KPFM measurements before the PFM measurements in order to offset the surface potential, and ideally afterwards as well to make sure the measurement is valid. Without this, the PFM measurements remain suspect (see Scientific Reports volume 7, Article number: 41657 (2017)). It is unclear how many SnS flakes were measured by PFM - there does not seem to be any statistics. How reproducible are these results? Also, the allocation of polar directions seems a bit arbitrary - how are the 1 and 3 directions allocated? This has to do with how the lattice is aligned, and PFM on random flakes cannot provide this information. Perhaps, PFM carried out by scanning along different directions might shed some light, but without this information, there is not much meaning to the calculated piezoelectric coefficients. I highly recommend the authors to refer to PFM studies on single-atomic layer MoS₂ [Nature volume 514, pages 470–474 (2014)] as an excellent example of how PFM should be conducted and interpreted for 2d materials.

It is also not clear why SnS should have an "enhanced" d₁₁ coefficient - I am not sure this is indeed the measured coefficient as the indices seem to have been designated arbitrarily (see previous point). Also, ideally, the PFM measurements should be carried out with the flakes in direct contact with a conducting substrate, and not SiO₂. This introduces other complex interface effects that have not been accounted for.

I am also concerned by the generator experiments themselves. The manuscript says that a

"tapping force" was applied, but there is no information about the magnitude. This should be provided to set the results in context. Also, it is not clear as to why power per unit length is of any importance in the case of these types of generators.

In conclusion, while the fabrication of these monolayers is interesting, I believe the main claims of the paper with respect to the piezoelectric properties and applications of SnS monolayers require further study and justification. Hence I am unable to recommend publication in Nature Communications.

Response to the reviewers' comments

The following is a point-by-point response to the reviewers' comments.

Reviewer 2

The authors addressed most of the questions I asked.

Our response

We are sincerely pleased to see your comment.

Reviewer's comment

1. One more question is about the growth map. From the growth map, can I conclude that the as-grown SnS film change to SnS₂ if the film exposes to H₂S for a time longer than ~17 min? Do this mean the as-grown SnS film is chemically unstable? I suggest the authors prepare a more accurate growth map.

Our response

Thanks for your critical question on the growth map.

The surface layer on the liquid metal, under specific H₂S concentration of this work at 375 °C at around 7 seconds, is a monolayer SnS. Longer time of exposure produces a combination of SnS and SnS₂. After ~ 17 seconds only SnS₂ is produced and this film becomes thicker with time thereafter. If any of the parameters, i.e. temperature or time, is changed the final product alters in stoichiometry.

However it is important to consider that, the films are stable after the delamination and no further change in thickness and stoichiometry is seen. Longer exposure of the delaminated SnS sheets to H₂S does not cause any more reactions as there is no available tin precursor. In fact, delamination of the sheet onto the substrate stops the reaction.

To prepare SnS or SnS₂, we must grow them on the droplet surface before we delaminate the sheet, which is controlled by the temperature limit of 375 °C or the time. If any of these increases, the product would likely be SnS₂ otherwise, by the tight control of these physical

parameters, stable monolayer SnS can be obtained confidently, as schematically shown in Figure S5.

If this SnS layer is not delaminated, as time passes, SnS₂ islands on top of the SnS thin film form due to the continued exposure to the precursors in the interface. If this surface reaction continues to progress, eventually the islands will meet up, establishing an intermediary layer of SnS₂. We limit further comments on the quality or properties of this SnS₂ layer as it is out of the focus of this paper. The second factor is the temperature at which the droplet is heated. If it is above 425°C, the conditions tend to result in a faster reaction with the sulfur source in the ambient environment and hence the material grown on the surface of the droplet is full stoichiometric SnS₂.

Again, we have to emphasize that the reaction stops after the delamination takes place, as no more tin will be available.

To address your comment, we have added the following Supplementary Note to the Supplementary Information and have edited the time axis (y-axis) from 'Time before transfer' to 'Time between preconditioning and delamination'.

Supplementary Note S2. Discussion on the growth map illustrating the link between growth conditions and products of liquid metal based-sulphide synthesis.

“Several factors are involved in the stoichiometry of the obtained tin sulphide films: temperature, and the time between the preconditioning step (when we remove a surface layer to obtain a clean surface) and film delamination and placement on a substrate. This time period is termed as the ‘transfer time’ and is measured in seconds. After the film is placed on a desired substrate, the reaction stops as there would be no precursor tin in liquid metal form available.

At the critical temperature of 375°C, and for the H₂S concentration conditions presented in the Methods section, the grown material on the surface of the liquid metal droplet is initially a thin SnS monolayer. It forms after the self-limiting reaction between the surface of liquid metal Sn and the sulfur-rich gaseous environment. At 375°C, this monolayer SnS can be harvested at under ~7 seconds. If this formed layer is not removed, SnS₂ islands will be gradually established on top of the SnS thin film due to the continued exposure to H₂S. If this

surface reaction progresses further, eventually the islands will meet up and form a top layer of SnS₂. However, at >17 seconds SnS₂ becomes very thick and is the dominating material within the film and no SnS can be practically detected. The second factor is the temperature at which the droplet is heated. At 250°C to 375°C, monolayer SnS, a combination of SnS and SnS₂ islands and then eventually SnS₂ films are obtained depending on the waited time. However, when it is above 375°C, the conditions are conducive for a faster reaction with the sulfur source in the ambient environment and hence the material grown on the surface of the droplet is SnS₂. A growth map is illustrated in **Fig. S5**, highlighting the important time and temperature parameters for the synthesis process. It is important to note that the layer reaction stops the moment when it is delaminated and transferred onto a substrate. The transferred film on a substrate does not change its stoichiometry or thickness. In the monolayer SnS case, it remains a stable homogenous monolayer on the surface in the normal ambient condition.”

Fig. S5 Growth map illustrating the link between growth conditions and the products. Inset into each region is TEM image and SAED pattern, respectively. The SnS₂ layer can be indexed to hexagonal Berndtite-4H SnS₂ (PDF card no: 21-1231).

Reviewer's comment

2. The other question is still about the morphology of the SnS film. Optical images (the updated images) can only give poor contrast of the film. High-quality SEM images are required to prove the overall quality of the transferred SnS film. I suggest to authors present a group (array) of SEM images of the SnS film with a size of few tens micrometers by a few tens micrometers.

Our response

Thank you for your constructive suggestion about presenting the quality of as-prepared SnS sheets. As you suggested, we took sequential high quality field emission SEM of a large SnS flake in order to provide an array of images of the sheets across a sample. We show two different sheets. Please see the Figure S3 below which is also added to the Supplementary Information. The SEM images clearly show the high quality and continuity of the sheets to several hundreds of micrometers to further support the validity of the optical images.

The following line was added to the Main text :

“ Further analysis by high-resolution scanning electron microscopy (**Fig. S3**) shows continuous laterally large sheets with no pinholes.”

The following was added to the Methods section:

“HR-SEM imaging was performed via FEI Verios model 460L using through the lens low energy detector operating at 950 V.”

Sheet 1

Sheet 2

Fig. S3 Array of FESEM images of the SnS monolayer, indicating the homogeneity and high quality of the sheets presented as two examples of: a) Sheet 1 b) Sheet 2. Each area is $50 \mu\text{m} \times 50 \mu\text{m}$.

Reviewer 3 (Remarks to the Author):

The manuscript by Khan et al. reports on liquid metal-based synthesis of monolayer SnS, and their resulting piezoelectric properties. The authors have made changes based on reviewers' comments, however, there still remains significant issues. With regards the fabrication process itself, previous reviewers have raised concerns which the authors seem to have adequately addressed in their revised version. However, I am still not convinced about the validity of the measurements of piezoelectric properties, and here I feel there are major gaps in the understanding of the phenomenon. The fact that data related to the output of the interdigitated electrode geometry had to be removed without explanation raises severe concerns. It is highly unsatisfactory to simply remove data that was put forward as evidence of piezoelectricity, without explaining what the measured signals were in the first place. This, unfortunately raises doubts about subsequent measurements and interpretation. Also, the authors seem to have confused the piezoelectric and piezotronic effects, which are distinct effects in semiconductors, and there is insufficient discussion of the latter in this respect.

Our response

We respect the comments. However, these claims are inaccurate. We have not removed any previous measurements but just moved them to the Supporting Information as had been suggested by the reviewer. Our usage of the terms such as piezotronic and piezoelectric are correct. Additionally, the main paper and Supporting Information are now very long and as such, adding more discussion that can be found in previous literature, is not practical.

Reviewer's comment

I have concerns regarding the PFM measurements. In contact-mode operation, there is significant electrostatic contribution which can be considerable particularly for thin 2d materials. This effect has to be corrected for by performing KPFM measurements before the PFM measurements in order to offset the surface potential, and ideally afterwards as well to make sure the measurement is valid. Without this, the PFM measurements remain suspect (see Scientific Reports volume 7, Article number: 41657 (2017)).

Our response

We addressed the comment. Per the suggestion we have conducted KPFM and performed the extra PFM to cater for the electrostatic contributions. Figure S13 below shows the KPFM of the flake before and after a PFM measurement.

The following was added to the Main text:

“KPFM measurements were performed prior to the PFM measurements to cater for any electrostatic charge contributions to the output signal¹ and were performed again after the measurements to validate the data further (**Fig S13.** and **S14**). Further discussion is found in the Supporting Information.”

We have also added the following to the Methods section

“KPFM was conducted prior to, and following, PFM imaging using the same cantilever (Electrolever) operating in AC mode and NAP mode (at a distance of 50 nm from the surface). The surface was electrically grounded to the AFM stage. In this method of imaging, the surface is first scanned with the cantilever in contact with the substrate (AC mode) and then a second pass of the same scan line is conducted at a distance of 50 nm from the surface. Here, the second pass of the surface produces a surface potential image of the surface. This provides a measure of the surface potential of the scanned area (the SnS flake and the substrate).”

Fig. S13 KPFM measurements for determination of surface potential a) Topography of SnS monolayer b) KPFM measurement before the PFM measurement c) KPFM measurement after the PFM measurement d) Surface potential histograms indicating negligible surface potential of 15 mV before and 30 mV after the contact mode PFM analysis. There is a negligible surface potential of 15 mV initially as measured. After the PFM measurements, there is negligible residual charge build-up, which is not particularly over the flake itself and is now 30 mV.

Another indication that surface charge is minimal and not of major concern in these measurements is the scan at 0 volt driving amplitude (**Figure S14**). Another consistent precaution in all measurements performed for the PFM was the loading of the sample on a silver chuck with silver paste to ensure the sample is grounded and thus surface charge is minimal.

Fig. S14 PFM measurements a) Topography of the SnS monolayer b) Amplitude response when applying 0 V driving voltage depicting no response at 0 V driving amplitude. These measurements are another indication that surface charge is minimal and not of a major concern in these measurements. Another consistent precaution in all measurements performed for the PFM was the loading of the sample on a silver chuck with silver paste. A connection line of silver paste is also drawn from the flake to the chuck to ensure the sample is grounded and thus surface charge is minimal.

Reviewer's comment

It is unclear how many SnS flakes were measured by PFM - there does not seem to be any statistics. How reproducible are these results?

Our response

To address this comment, the PFM measurements were performed on three different flakes and an error bar has been added to show the deviations in the measurements with the average value across each sheet being used to plot the actual amplitude for each driving voltage.

Yes, the results are reproducible as shown by the small average deviation in amplitudes (± 0.3 pm/V). The samples were used several days after synthesis and for many consecutive measurements with the same result showing the material is stable and results are hence reproducible. The updated Figure 4 in the main manuscript is shown below.

Fig. 4 Lateral piezoelectric response of the monolayer SnS a) Topography of the monolayer sheet with the height profile of the sheet illustrating it to be ~1 nm, which is approximately a monolayer. This is also confirmed by the histogram showing the height distribution b) Lateral amplitude of the SnS monolayer with inset of histogram c) Lateral phase of the SnS monolayer with inset of histogram d) Lateral piezoresponse of monolayer SnS and PPLN vs the driving voltage. Scale bar is 2 μm .

The coefficient calculation has been updated accordingly and the following text was added to the Main text to reflect these additional measurements:

“The plot is fitted to a linear line, which gives $1.2558 \text{ a.u.V}^{-1}$ and $0.2864 \text{ a.u.V}^{-1}$ for the monolayer SnS and the PPLN sample, respectively. Because $d_{\text{eff}} = 5.95 \text{ pmV}^{-1}$ for the PPLN in the lateral direction (**Fig S15**), an effective piezoelectric coefficient of $26.1 \pm 0.3 \text{ pmV}^{-1}$ is obtained for monolayer SnS.”

Reviewer’s comment

Also, the allocation of polar directions seems a bit arbitrary - how are the 1 and 3 directions allocated? This has to do with how the lattice is aligned, and PFM on random flakes cannot provide this information. Perhaps, PFM carried out by scanning along different directions might shed some light, but without this information, there is not much meaning to the calculated piezoelectric coefficients. I highly recommend the authors to refer to PFM studies on single-atomic layer MoS₂ [Nature volume 514, pages470–474(2014)] as an excellent example of how PFM should be conducted and interpreted for 2d materials.

Our response

In terms of the assignment of the coefficient directions, we did not assign our coefficient any specific directions and have called it a lateral or in-plane coefficient. When using the lateral piezoresponse force (LPFM) microscopy mode, the stimulus is provided from above the sheet (hence the out of plane (3)) and the response is measured in-plane. The calibration sample was measured in the exact same way.

We agree that a directional dependent of the PFM measurements would be beneficial to further understanding and this is a future direction we may take but would be a fair amount of additional work for this paper. Also, our lab does not have access to equipment that would allow us to determine the directions of the grown sheets. Collaborating further on this would jeopardize our ability to respond to the review in a timely manner. Our main intention of performing the PFM measurements was to provide further validation for the piezoelectric property of the material being used in the nanogenerator fabrication.

Reviewer's comment

It is also not clear why SnS should have an "enhanced" d_{11} coefficient - I am not sure this is indeed the measured coefficient as the indices seem to have been designated arbitrarily (see previous point). Also, ideally, the PFM measurements should be carried out with the flakes in direct contact with a conducting substrate, and not SiO₂. This introduces other complex interface effects that have not been accounted for.

Our response

We feel the use of the term 'enhanced' piezoelectric coefficient may have been misleading to the reviewers. What we meant by 'enhanced' piezoelectric coefficient was based on the intrinsic structure of SnS rather than the measured coefficients being enhanced due to our work or experiments. Rather, we expect this superior piezoelectricity due to the puckered lattice structure providing flexibility in the lateral directions. Hence, we have changed the term 'enhanced' to 'giant' and modified the text to the following to ensure readers also understand that this is an inherent property of the material and not enhanced due to our work. The use of a SiO₂/Si for PFM measurements has been reported previously^{2,3,4} and may not be as relevant in lateral measurements in comparison to the conventional vertical PFM measurements.

Modified sentence as in Main text:

"This difference can be associated to the giant d_{11} value for monolayer SnS as per DFT calculations in comparison to that of MoS₂ and WSe₂ that are less than 4 pmV⁻¹ in majority of past reports"

Reviewer's comment

I am also concerned by the generator experiments themselves. The manuscript says that a "tapping force" was applied, but there is no information about the magnitude. This should be provided to set the results in context.

Our response

We have edited the sentence in the Main text to read as below with the tapping force stated.

“The first mode of testing was the d_{31} mode, where force was applied normal to the-plane direction by tapping the device surface in a controlled manner with a measured peak-to-peak load amplitude of ~ 4 N and in the frequency range of 1-10 Hz (**Fig. S17**).”

We have also added additional testing information to the Methods section.

The impactor head, that was used to provide a tapping action, was homed to the surface of the device as the initial setpoint (i.e $d_{\min} = 0$ mm) with a low pre-load of ~ 3 N. A square pulse of set amplitude (Δd) of 2 mm and frequency of a desired number of cycles was then applied to the device surface where displacement of the head is adjusted to d_{\max} of ~ 2 mm. The tests were done at an R-ratio of 0.43 where the minimum and maximum load was measured to be ~ 3 N and ~ 7 N, respectively. The maximum contact load and maximum displacement were recorded via data acquisition software.

Reviewer's comment

Also, it is not clear as to why power per unit length is of any importance in the case of these types of generators.

In conclusion, while the fabrication of these monolayers is interesting, I believe the main claims of the paper with respect to the piezoelectric properties and applications of SnS monolayers require further study and justification. Hence I am unable to recommend publication in Nature Communications.

Our response

We highlighted the power/unit length as an important parameter since our method can achieve large laterally sized sheets. In general, measurements have been carried out of materials fabricated by applying other conventional synthesis methods where the flake is of a few nanometers in lateral dimensions whereas with our method we can achieve large scales.

Since research is about innovation and new science, we feel that this newly introduced parameter is more adequate at quantifying the advantages of such systems and we hope that it will be adopted as a new way to measure the performances of such systems.

References

- 1 Kim, S., Seol, D., Lu, X., Alexe, M. & Kim, Y. Electrostatic-free piezoresponse force microscopy. *Sci. Rep.* **7**, 41657 (2017).
- 2 Wang, X. *et al.* Subatomic deformation driven by vertical piezoelectricity from CdS ultrathin films. *Science Advances* **2**, e1600209 (2016).
- 3 Syed, N. *et al.* Printing two-dimensional gallium phosphate out of liquid metal. *Nat. Commun.* **9**, 3618 (2018).
- 4 Nasr Esfahani, E., Li, T., Huang, B., Xu, X. & Li, J. Piezoelectricity of atomically thin WSe₂ via laterally excited scanning probe microscopy. *Nano Energy* **52**, 117-122 (2018).

Reviewers' comments:

Reviewer #2 (Remarks to the Author):

All my questions were well addressed, I would like to recommend publication in Nature Communications.

Reviewer #3 (Remarks to the Author):

The authors have addressed most of the points raised in the previous round of review, and in principle, the manuscript is acceptable. However, the authors may want to reconsider the necessity of including the section on "piezotronic" effects in this material, when they do not actually measure this at all. In order to demonstrate the modulation of barrier height (as defined by the piezotronic effect), they would need to show current-voltage data to show how strain affects barrier height. This is not necessary to the conclusions presented in the paper, which relate to the piezoelectric effect in this material. I find this part confusing as a reader, and cannot see why it is being highlighted if the effect is not actually being measured/demonstrated.

Also, I am not convinced that "power/unit length" is an effective parameter, and I am not sure why the authors feel so strongly about this. They mention this in the abstract stating that it is highest in their material, but in the text they only compare it to one other published report. Hence I fail to see the significance. I think the results are quite significant as they are without the need to introduce new parameters into the literature which have a limited purpose. Perhaps they can justify the need for this by presenting a table comparing with a larger number of reports.

Lastly, it is understandable that the authors are unable to provide a detailed PFM study to fully address the question about the allocation of directions, but the justification for their choice of indices should be stated in the text to avoid confusion to readers, just as they have explained in the response letter.

I would recommend that the above (minor) points be addressed in the manuscript prior to publication.

Reviewer #4 (Remarks to the Author):

The manuscript by Khan et al. has presented a new technique for synthesis of SnS 2D materials. They have demonstrated the piezoelectric property of SnS by two different measurement techniques. Exploring piezoelectricity in 2D materials add a new facet in studying its novel applications. My comments about the paper are listed below:

1. The original report WS₂ by Wu et al. illustrate the piezoelectric dependence of odd layers and even layers with considering the projected symmetry. For odd layers such as 1, 3 and 5, WS₂ has piezoelectricity, but for even layers such as 2 and 4 layers, there is no piezoelectricity. As for the current work, SnS, have the authors observed such phenomenon?
2. The manuscript mainly present output voltage of the nanogenerator, but there is no data presented about the output current. Both current and voltage signals are required to prove that the nanogenerator gives out power.
3. The number of figures in the text are too many and they can be combined, fig. 1+ fig. 2 can be one figure.

Response to the reviewers' comments

We would like to thank the reviewers for their insightful comments and their valuable time spent on the manuscript. The following is a point-by-point response to the reviewers' comments.

Reviewer 2 (Remarks to the Author):

All my questions were well addressed; I would like to recommend publication in Nature Communications.

Our response

We are sincerely pleased to see your comment and thank you for your perceptive comments which helped us to improve the quality of our manuscript.

Reviewer 3 (Remarks to the Author):

The authors have addressed most of the points raised in the previous round of review, and in principle, the manuscript is acceptable.

Our response

We are happy to see we have addressed most of the points from the previous round to your satisfaction and we appreciate your further inputs.

Reviewer's comment

However, the authors may want to reconsider the necessity of including the section on "piezotronic" effects in this material, when they do not actually measure this at all. In order to demonstrate the modulation of barrier height (as defined by the piezotronic effect), they would need to show current-voltage data to show how strain affects barrier height. This is not necessary to the conclusions presented in the paper, which relate to the piezoelectric effect in this material. I find this part confusing as a reader, and cannot see why it is being highlighted if the effect is not actually being measured/demonstrated.

Our response

Thanks for the comment. We understand your point of view and hence as per your suggestion have removed the following from the Main text since it does not change the conclusions of the work:

“Here a modulation of electronic transport occurs across the interface with the input of dynamic mechanical stimulation, resulting in the redistribution of charges, which alters the band structure near the interface. The Schottky barrier, which is a distinct discontinuity in energy levels at the interface, exists between metal electrodes (Cr/Au) and SnS monolayers. Upon straining, the remnant piezoelectric polarization generated at the SnS monolayer side can significantly impact the Schottky barrier height and the barrier

interface is depleted of major carriers, which in the case of *p*-type SnS are holes. The local Schottky barrier is depleted by the piezoelectric polarization and becomes less depleted, resulting in a decreased Schottky barrier height. Therefore, strain-induced polarization can effectively modulate the electronic transport across the metal-semiconductor contact.”

Reviewer's comment

Also, I am not convinced that "power/unit length" is an effective parameter, and I am not sure why the authors feel so strongly about this. They mention this in the abstract stating that it is highest in their material, but in the text they only compare it to one other published report. Hence I fail to see the significance. I think the results are quite significant as they are without the need to introduce new parameters into the literature which have a limited purpose. Perhaps they can justify the need for this by presenting a table comparing with a larger number of reports.

Our response

Thank you for this comment. We appreciate that the readers may wish to see a more conventional parameter for an easier comparison. Hence, taking your feedback on board, we have done the following:

- 1) We have developed a new graph of comparison in **Figure 5b** which shows comparative reports with voltage output and the more conventional power density parameter.
- 2) The table with the corresponding values for power density and other relevant parameters of comparison are shown in the Supplementary Information in **Table S2** for all the reports.
- 3) We have updated our discussion on power density in the text.

We hope this addresses your comment and the amended text in the Main file of the manuscript is as follows:

“Power density of the devices was calculated by taking the ratio of the peak voltage power and the area of the devices. The highest previous report on power density of 2D nanogenerator devices has been reported for MoS₂ (2 mWm⁻²)¹, while our nanogenerator shows a significantly higher power density of ~24 mWm⁻². Fig. 5b and Table S2 summarises the parameters and compares our work with previous reports published on nanogenerators which are based on 2D materials highlighting their key performance criteria including the output voltage and power density.”

Fig. 5. The output measurements of the device with synthesised monolayer SnS on a flexible substrate in d_{11} mode, for practical wearable device applications: a) Voltage

output of tensile bending and relaxing mode action as depicted in the inset (two-electrode device), b) a graph showing a comparison of voltage and **power density outputs** of previous 2D-based piezoelectric nanogenerators ¹⁻⁴ reported in comparison to our work in normalised strain.

Table S2. Table of important parameters of previous reports on 2D material-based nanogenerators

2D Material	Parameters					Ref	Piezoelectric coefficients	
	V	I	Maximum instantaneous power @ stated load resistance	Test conditions	Power density		Theoretical	Experimental
	Units							
	mV	pA	pW	% strain	mWm ⁻²		pmV ⁻¹	
Monolayer-MoS ₂	18 (peak OC voltage)	27 (peak SC current)		0.64		1	$d_{11}=3.65^5$ $d_{11}=4.94^6$ $d_{11}=3.73^7$	
	3.5 (calc)	12.5 (calc)	55.3 fW (220 MΩ)	0.53	2			
Monolayer-MoS ₂ (armchair direction)	20 (peak output)	30 (peak output)		0.48, 0.5 Hz, 70mm		4	$d_{11}=3.65^5$ $d_{11}=4.94^6$ $d_{11}=3.73^7$	$d_{11}=3.78$ (AC) ⁴
Monolayer-MoS ₂ (zigzag direction)	10 (peak output)	20 (peak output)						$d_{11}=1.38$ (ZZ) ⁴

Monolayer-MoS ₂ (armchair direction)	20 (1 GΩ) graph		0.4 (1 GΩ)-graph		0.5			
Monolayer MoS ₂ (zigzag direction)	10 mV (1 GΩ) graph		0.2 (1 GΩ)-graph		0.25			
Monolayer-WSe ₂	~35 (calc)	72 (calc)	2.54 (500 MΩ)	0.39	0.5	2	$d_{11}=2.79$	$d_{11}=3.26 \pm 0.3$ ²
Turbostratic-WSe ₂	~57 (graph)	71 (calc)	4.05 (~ 0.1 GΩ)	0.89	0.8		$d_{11}=0-1.5$ ²	
Monolayer-WSe ₂	~37 (graph)	55 (calc)	2.05 (~ 0.5 GΩ)	0.89	0.4			
Pristine MoS ₂	10 (1GΩ)	30 (output) 1GΩ	0.07 (500 MΩ)	0.48	0.007	3	$d_{11}=3.65$ ⁵ $d_{11}=4.94$ ⁶ $d_{11}=3.73$ ⁷	$d_{11}=3.06 \pm 0.6$ ³
S-treated MoS ₂	20 (1GΩ)	100 (output) 1GΩ	0.73 (500 MΩ)	0.48	0.073		$d_{11}=3.73 \pm 0.2$ ³	
Monolayer-SnS	~150 (1GΩ)	160 (output) (1GΩ)	24 (1 GΩ)	0.70	24	This work	$d_{11}=144.76$ ⁸ $d_{12}= -22$ ⁸	PFM measured value: 26.1 ± 0.3

Note: Values in green were calculated or interpreted by the authors. AC- armchair, ZZ- zigzag

Reviewer's comment

Lastly, it is understandable that the authors are unable to provide a detailed PFM study to fully address the question about the allocation of directions, but the justification for their choice of indices should be stated in the text to avoid confusion to readers, just as they have explained in the response letter.

Our response

Thank you for this recommendation of explaining the choice of indices better in the manuscript to avoid confusing the readers.

We have done this as per your suggestion by adding the following text to the Main text:

“When using LPFM microscopy mode, the stimulus is provided from above the sheet (hence out of plane) and the response is measured in-plane. The calibration sample was measured in the exact same way.”

Reviewer's comment

I would recommend that the above (minor) points be addressed in the manuscript prior to publication.

Our response

We hope that we have addressed all the points to your satisfaction and that the work is now suitable for publication.

Reviewer 4 (Remarks to the Author):

The manuscript by Khan et al. has presented a new technique for synthesis of SnS 2D materials. They have demonstrated the piezoelectric property of SnS by two different measurement techniques. Exploring piezoelectricity in 2D materials add a new facet in studying its novel applications.

Our response

Thank you for the time in reviewing the manuscript and appreciating our work. We also thank you for your intuitive comments which we intend to address carefully below.

My comments about the paper are listed below:

Reviewer's comment

1. The original report WS₂ by Wu et al. illustrate the piezoelectric dependence of odd layers and even layers with considering the projected symmetry. For odd layers such as 1, 3 and 5, WS₂ has piezoelectricity, but for even layers such as 2 and 4 layers, there is no piezoelectricity. As for the current work, SnS, have the authors observed such phenomenon?

Our response

Thank you for your interesting comment. This layer dependency phenomenon may exist in theory if we draw a similar conclusion about the trend theoretically calculated in another Group IV monochalcogenide, SnSe⁹.

However, this has never been proven experimentally in the past 5 years and we believe the reason is as we described in the paper "*To make the matter more challenging, the formation of high quality monolayers of group IV monochalcogenides, specifically SnS, using conventional exfoliation, has been suggested to be limited due to the strong inter-layer interactions by the lone-pair electrons of S, which are much stronger than the van der Waals forces between the layers*"^{10,11},

This strong inter-layer coupling by the lone-pair electrons of S renders conventional methods such as micromechanical cleaving futile in producing the monolayer or multilayer structure

and the whole novelty of our paper is the unprecedented synthesis of monolayer crystalline SnS. The liquid metal synthesis approach, we have established and demonstrated, so far is the only practical way to make the 'monolayer' of this material (even with our method the second layer appears as SnS₂ as presented in the Main texts and Supplementary Information – **Supplementary Note S2** and **Fig S5**). Also, the largest piezoelectric effect is expected in the monolayer⁸ and consequent layer outputs would be either non-existent or much less.

Reviewer's comment

2. The manuscript mainly present output voltage of the nanogenerator, but there is no data presented about the output current. Both current and voltage signals are required to prove that the nanogenerator gives out power.

Our response

Thanks for the comment. As per your accurate suggestion, we have added the following figure in the Supplementary Information:

Fig. S19. Current output of the SnS monolayer device: a) Test setup for current measurement with inset of the device optical image b) Current output waveform of the device tested with 1 GΩ resistance when a force was applied at t=1 s and released at t=3 s.

We have added the following in the Main text:

“Current measurements were also done with $1\text{G}\Omega$ passive resistance (**Fig. S19**) which produced ~ 160 pA output current.”

We have also added the following to the Methods section:

“Current data was measured using a KEYSIGHT B2912A precision source meter capable of measuring in 10 fA range. Data was acquired every 0.1 s using a Labview program connected to the system in an electromagnetic shielding chamber. When testing the device, it was connected to an external resistance of $1\text{G}\Omega$. The same strain of approximately 0.7% was applied using an automated probe.”

Reviewer's comment

3. The number of figures in the text are too many and they can be combined, fig. 1+ fig. 2 can be one figure.

Our response

We agree with your valid comment and hence have reduced the number of figures by combining **Fig. 1** and **2**. Please see the amended figure and relevant caption below:

Fig. 1. Synthesis schematic representation and material characterisations of the 2D SnS:

a) Schematic depiction of the synthesis process, monolayer of SnS and its application on a nanogenerator transducer. b) TEM image of monolayered SnS nanosheet c) HRTEM fringe pattern showing d -spacing of 0.29 nm matched to the plane (101) with inset that shows SAED pattern also indicating a 0.29 nm d -spacing matched to the (101) plane d) dark field TEM image of a 2D SnS that shows very low presence of grain boundaries, confirming near single crystallinity of the planes. e) Raman spectrum showing the 4 characteristic Raman active vibration modes that are associated with the A_g modes and the B_{3g} mode. Four characteristic Raman active modes at 96, 190 and 220 cm^{-1} are associated with the A_g modes, while a peak at 163 cm^{-1} assigned to its B_{3g} mode are expected in bulk SnS. f) Thickness profile showing sheet thickness of ~ 0.7 nm, which can be matched to one-unit cell thickness of monolayer SnS and g) the corresponding AFM of the SnS monolayer.

References

- 1 Wu, W. *et al.* Piezoelectricity of single-atomic-layer MoS₂ for energy conversion and piezotronics. *Nature* **514**, 470 (2014).
- 2 Lee, J.-H. *et al.* Reliable piezoelectricity in bilayer WSe₂ for piezoelectric nanogenerators. *Adv. Mater.* **29**, 1606667 (2017).
- 3 Han, S. A. *et al.* Point-defect-passivated MoS₂ nanosheet-based high performance piezoelectric nanogenerator. *Adv. Mater.* **30**, 1800342 (2018).
- 4 Kim, S. K. *et al.* Directional dependent piezoelectric effect in CVD grown monolayer MoS₂ for flexible piezoelectric nanogenerators. *Nano Energy* **22**, 483-489 (2016).
- 5 Blonsky, M. N., Zhuang, H. L., Singh, A. K. & Hennig, R. G. Ab Initio Prediction of Piezoelectricity in Two-Dimensional Materials. *ACS Nano* **9**, 9885-9891 (2015).
- 6 Alyörük, M. M., Aierken, Y., Çakır, D., Peeters, F. M. & Sevik, C. Promising piezoelectric performance of single layer transition-metal dichalcogenides and dioxides. *J. Phys. Chem. C* **119**, 23231-23237 (2015).
- 7 Duerloo, K.-A. N., Ong, M. T. & Reed, E. J. Intrinsic piezoelectricity in two-dimensional materials. *J. Phys. Chem. Lett.* **3**, 2871-2876 (2012).
- 8 Fei, R., Li, W., Li, J. & Yang, L. Giant piezoelectricity of monolayer group IV monochalcogenides: SnSe, SnS, GeSe, and GeS. *Appl. Phys. Lett.* **107**, 173104 (2015).
- 9 Fang, W. *et al.* Layer dependence of geometric, electronic and piezoelectric properties of SnSe. *Preprint at <https://arxiv.org/abs/1603.01791>* (2016).
- 10 Lefebvre, I., Szymanski, M. A., Olivier-Fourcade, J. & Jumas, J. C. Electronic structure of tin monochalcogenides from SnO to SnTe. *Phys. Rev. B* **58**, 1896-1906 (1998).
- 11 Higashitarumizu, N. *et al.* Self-passivated ultra-thin SnS layers via mechanical exfoliation and post-oxidation. *Nanoscale* **10**, 22474-22483 (2018).

REVIEWERS' COMMENTS:

Reviewer #3 (Remarks to the Author):

I am satisfied with the changes made to the manuscript and recommend its acceptance.

Reviewer #4 (Remarks to the Author):

The authors have fully responded to my questions and the paper is well revised. It is ready for publishing.